# FRBNet: Revisiting Low-Light Vision through Frequency-Domain Radial Basis Network

**Fangtong Sun**[*], **Congyu Li**[*], **Ke Yang, Yuchen Pan, Hanwen Yu, Xichuan Zhang**[†], **Yiying Li**[†]

Intelligent Game and Decision Lab (IGDL), Beijing, China
{sunfangtong19, liyiying10}@nudt.edu.cn
licongyu@hnu.edu.cn, zhxc@alu.hit.edu.cn

## Abstract

Low-light vision remains a fundamental challenge in computer vision due to severe illumination degradation, which significantly affects the performance of downstream tasks such as detection and segmentation. While recent state-of-the-art methods have improved performance through invariant feature learning modules, they still fall short due to incomplete modeling of low-light conditions. Therefore, we revisit low-light image formation and extend the classical Lambertian model to better characterize low-light conditions. By shifting our analysis to the frequency domain, we theoretically prove that the frequency-domain channel ratio can be leveraged to extract illumination-invariant features via a structured filtering process. We then propose a novel and end-to-end trainable module named **F**requency-domain **R**adial **B**asis **Net**work (**FRBNet**), which integrates the frequency-domain channel ratio operation with a learnable frequency domain filter for the overall illumination-invariant feature enhancement. As a plug-and-play module, FRBNet can be integrated into existing networks for low-light downstream tasks without modifying loss functions. Extensive experiments across various downstream tasks demonstrate that FRBNet achieves superior performance, including +2.2 mAP for dark object detection and +2.9 mIoU for nighttime segmentation. Code is available at: https://github.com/Sing-Forevet/FRBNet.

## 1 Introduction

In recent years, computer vision tasks such as object detection [28] and semantic segmentation [39] have achieved remarkable progress, driven by advances in deep learning techniques [19, 54, 57] and the availability of large-scale annotated datasets [33, 14, 69, 7]. The models underlying these tasks are typically trained on well-lit, high-quality images [37, 48], which often suffer from significant performance degradation when employed under low-light conditions. Moreover, available real-world low-light datasets [40, 73] remain relatively small in scale, hindering effective low-light network training.

To deal with low-light vision tasks, there are several mainstream methods: (1) Image enhancement methods, (2) Synthetic data training, (3) Multi-task learning strategies, and (4) Plug-and-play modules, as illustrated in Fig. 1 (a). Image enhancement aims to restore visual quality before feeding images into downstream models. It can enhance visibility for humans but may not guarantee machine perception performance [64]. Synthetic data methods[1, 13] address low-light data scarcity via image signal processes and other techniques, such as Dark ISP[10], but face high costs, limited diversity, and realism gaps. Multi-task learning jointly optimizes multiple objectives via complex loss functions

---

[*]Equal Contributions. [†] Corresponding Authors.

39th Conference on Neural Information Processing Systems (NeurIPS 2025).

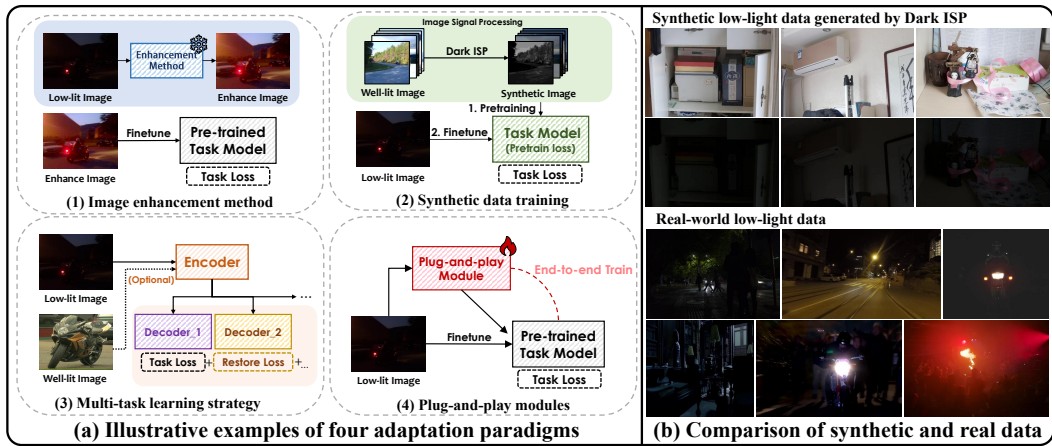

Figure 1: (a) Illustrative examples of four adaptation paradigms for low-light vision tasks, and (b) Comparison between synthetic low-light data (top) and real-world low-light data (bottom), demonstrating the higher complexity of real-world scenarios with localized light sources and non-uniform illumination patterns that synthetic methods struggle to accurately simulate.

but faces optimization challenges on large solution spaces. Unlike these, plug-and-play paradigms to enhance illumination-invariant features, such as DENet [46] and PE-YOLO [74], gain attention for their high applicability and flexibility to adapt to various basic network architectures.

For plug-and-play paradigms, FeatEnHancer[21] improves low-light vision tasks via a hierarchical feature enhancement module. Subsequently, YOLA[5] employs zero-mean convolution to extract illumination-invariant features and achieves competitive performance. However, these methods lack a complete modeling of low-light images in the real world, some of which are based on incomplete assumptions such as the basic Lambertian model[29]. In addition, these spatial-domain convolution-based methods fall short in global perception due to fixed receptive fields.

Therefore, we revisit the imaging formation model and propose a plug-and-play module, termed Frequency-domain Radial Basis Network (FRBNet), for diverse low-light downstream tasks. Specifically, inspired by the Phong illumination model[45], we theoretically extend the classical Lambertian formulation[29] and construct an extended generalized low-light model. Due to the limitations of the spatial-domain channel ratio, we propose a frequency-domain channel ratio and a learnable frequency-domain filter based on optimized radial basis functions for illumination-invariant feature extraction. Modulated by a zero-Direct Current (zero-DC) Gaussian frequency window and orientation angle, this filter forms an overall lightweight plug-and-play module for frequency suppression and structure filtering. Extensive experiments on four representative low-light vision tasks: object detection, face detection, semantic segmentation, and instance segmentation demonstrate that FRBNet significantly surpasses baselines and achieves superior performance.

The main contributions of this paper can be summarized as follows.

- We theoretically extend the Lambertian model for real-world low-light conditions and then formulate the novel Frequency-domain Channel Ratio (FCR) for illumination-invariant feature enhancement. To the best of our knowledge, this is the first work that operates the channel ratio for illumination-invariant features in the frequency domain.

- We design a Learnable Frequency-domain Filter (LFF) with a zero-DC frequency window and an improved radial basis filter for robust feature extraction. This filter can process unde-sired frequency components adaptively by frequency suppression and angular modulation.

- Based on the theoretical analysis, we propose a lightweight plug-and-play module called Frequency-domain Radial Basis Network (FRBNet) which can be seamlessly integrated into various low-light vision tasks. It provides a frequency-domain illumination-invariant feature enhancement paradigm through the inter-relationships of channels constructed by FCR and the effective filtering by LFF. Comprehensive evaluations demonstrate that FRBNet outperforms existing state-of-the-art methods on various low-light vision downstream tasks.

## 2 Related work

### 2.1 Low-light vision for downstream tasks

Beyond direct image enhancement approaches[34, 62, 77, 49, 72, 31], recent research has explored alternative strategies for improving downstream vision tasks in low-light conditions. Several works leverage synthetic data generation to address the scarcity of real low-light datasets. DAINet [13] simulates low-light conditions through image signal processing to achieve zero-shot adaptation of detectors. WARLearn [1] uses unlabeled synthetic data to enhance representation learning for adverse weather robustness. Similarly, BrightVO [59] generates synthetic low-light data through CARLA [12] simulation to train brightness-guided Transformers for visual odometry tasks. Another paradigm involves joint enhancement and detection via multi-task learning[10, 9, 24, 27]. Recent benchmarks like RealUnify [53] explore if cross-task unified vision models consistently benefit performance. End-to-end optimization methods directly target downstream task performance rather than intermediate image quality [74, 38]. DENet [46] and FeatEnHancer [21] focus on feature-level enhancement through learnable modules integrated into detection networks. Subsequently, YOLA [5] extracts illumination-invariant features through channel-wise operations, directly improving detection performance in low-light conditions. We share the philosophy of end-to-end optimization; however, we note that existing approaches often overlook the complexity of real-world low-light scenarios, such as local light sources and uneven reflections, which are explicitly considered in our design.

### 2.2 Frequency-domain analysis in low-light image processing

Frequency-domain analysis has proven effective in low-light image enhancement [78, 23, 35] by separating illumination from structural details through spectral decomposition. Typically, low-frequency components represent global illumination and smooth variations, while high-frequency captures edges and textures [60]. FourLLIE [58] utilizes amplitude information to enhance brightness and recover details in a two-stage framework. Similarly, Frequency-Aware Network [52] selectively adjusts low-frequency components while preserving high-frequency details. Li *et al.*[32] employ frequency decomposition to guide hybrid representations for joint image denoising and enhancement. In the realm of generative models, FourierDiff[42] embeds Fourier priors into diffusion models for zero-shot enhancement and deblurring, while FCDiffusion [18] enables controllable generation through frequency band filtering. Beyond enhancement, FreqMamba [79] integrates frequency analysis with the Mamba architecture for effective image deraining. However, most existing frequency-domain approaches mainly operate at the pixel level by modifying low-frequency illumination components while preserving high-frequency details. In contrast, our method is the first to leverage channel ratio representations for extracting illumination-invariant features directly in the frequency domain, shifting the focus from pixel-level enhancement to feature-level learning.

## 3 Theoretical Analysis of Method Design

### 3.1 Extended generalized low-light model

The classical Lambertian image formation model [29, 51] characterizes low-light scenarios through the diffuse reflection assumption [8], expressing an image $I$ at pixel location $(x, y)$ as:

$$I_C(x, y) = m[\vec{n}(x, y), \vec{l}(x, y)] \cdot \varphi_C(x, y) \cdot \rho_C(x, y). \tag{1}$$

Here, $C \in \{R, G, B\}$ denotes the RGB color channel. $\vec{n}$ and $\vec{l}$ represent the surface normal and light direction, respectively. $m[\cdot, \cdot]$ is the interaction function, $\varphi_C$ denotes the illumination component, $\rho_C$ denotes the intrinsic reflectance component.

The Lambertian model assumes purely diffuse reflection, where light is scattered uniformly across the surface. However, real-world low-light images (Fig. 1(b)) frequently contain complex and spatially localized light sources, including streetlights, vehicle headlights, and neon signs. These sources contradict the idealized diffuse reflection assumption underlying the Lambertian model.

Motivated by the additive decomposition in the Phong illumination model [45]( A.1 for details), we introduce an extended version of the Lambertian model adapted to real-world low-light scenes by reinterpreting the localized light sources as non-uniform highlights, which can be expressed as:

$$I_C(x, y) = m[\vec{n}(x, y), \vec{l}(x, y)] \cdot \varphi_C(x, y) \cdot \rho_C(x, y) + S_C(x, y), \tag{2}$$

where $S_C$ represents a spatially irregular highlight component that can be further defined as:

$$S_C(x, y) = H_C(x, y) \cdot m[\vec{n}(x, y), \vec{l}(x, y)] \cdot \varphi_C(x, y) \cdot \rho_C(x, y), \tag{3}$$

with $H_C$ denoting the relative strength of highlight interference. For notational simplicity, we define $D_C(x, y) = m[\vec{n}(x, y), \vec{l}(x, y)] \cdot \varphi_C(x, y) \cdot \rho_C(x, y)$ as the standard diffuse reflection component. Substituting this into Eq. (2) and rearranging terms, we obtain a more concise expression:

$$I_C(x, y) = D_C(x, y) + S_C(x, y) = D_C(x, y) \cdot (1 + H_C(x, y)). \tag{4}$$

## 3.2 Frequency-domain channel ratio

Leveraging channel ratios (CR) to isolate illumination-invariant features has proven effective for low-light visual tasks [44, 17, 5]. Taking the channel ratio between the red channel $R$ and the green channel $G$ as an example, the log-transformed formulation, according to our extended generalized low-light model, can be obtained as:

$$\begin{aligned}
\mathrm{CR}_{RG} &= \log\left(\frac{I_R}{I_G}\right) = \log\left(\frac{\varphi_R \cdot \rho_R \cdot (1 + H_R)}{\varphi_G \cdot \rho_G \cdot (1 + H_G)}\right) \\
&= \log\varphi_R - \log\varphi_G + \log\rho_R - \log\rho_G + \log(1 + H_R) - \log(1 + H_G).
\end{aligned} \tag{5}$$

As shown in Eq. (5), the nonlinear residual from the highlight term disrupts the clean separation of illumination and reflectance, limiting the effectiveness of spatial-domain channel ratio methods. To overcome these limitations, we shift our analysis to the frequency domain, where illumination and reflectance components naturally occupy different frequency bands [60], enabling more effective separation of illumination-invariant features. Drawing inspiration from prior works on spatial-domain channel ratios [44, 17, 5], we innovatively propose the **F**requency-domain **C**hannel **R**atio (FCR) as:

$$\begin{aligned}
\mathrm{FCR}_{RG} &= \mathcal{F}[\log(\frac{I_R}{I_G})] \\
&= \mathcal{F}[\log\varphi_R - \log\varphi_G] + \mathcal{F}[\log\rho_R - \log\rho_G] + \mathcal{F}[\log(1 + H_R) - \log(1 + H_G)],
\end{aligned} \tag{6}$$

where $\mathcal{F}[\cdot]$ represents the Fourier transform operator. To handle the non-linear residual term $\Delta = \mathcal{F}[\log(1 + H_R) - \log(1 + H_G)]$, we apply a first-order Taylor expansion. Given that significant contributions in the data are usually sparse and localized, we assume that $H_C \in [0, 1)$ has a relatively small magnitude, allowing us to approximate $\log(1 + H_C)$ as $H_C + \mathcal{O}(H_C^2)$.

Under the aforementioned assumption, by neglecting higher-order terms, we can obtain a linearized approximation of $\Delta$ as follows:

$$\Delta = \mathcal{F}[H_R - H_G] = \mathcal{H}_R - \mathcal{H}_G, \tag{7}$$

where $\mathcal{H}_R$ and $\mathcal{H}_G$ denote the frequency-domain representations of $H_R$ and $H_G$, respectively. To investigate the spectral characteristics of the residual term $\Delta$, we decompose it into its amplitude and phase components:

$$\Delta = \mathcal{H}_R - \mathcal{H}_G = a_R \cdot e^{i\theta_R} - a_G \cdot e^{i\theta_G}, \tag{8}$$

where $a_R$, $a_G$ represent the amplitude terms, and $\theta_R$, $\theta_G$ denote the phase components. To characterize the phase relationship between channels, we introduce the frequency correlation coefficient $Cor_{RG} = e^{i(\theta_G - \theta_R)}$ (derived from A.2, see [56]), which quantifies the angular displacement between channel responses in the frequency domain. This allows us to reformulate $\Delta$ as:

$$\Delta = e^{i\theta_R} \cdot \left(a_R - a_G \cdot e^{i(\theta_G - \theta_R)}\right) = e^{i\theta_R} \cdot (a_R - a_G \cdot Cor_{RG}), \tag{9}$$

This factorization reveals that the residual term is structured as a phase-modulated component, where $e^{i\theta_R}$ serves as the carrier phase and $(a_R - a_G \cdot Cor_{RG})$ encodes the amplitude discrepancy modulated by the inter-channel phase correlation.

Finally, the ultimate formulation of the frequency-domain channel ratio can be summarized as:

$$\mathrm{FCR}_{RG} = \underbrace{\mathcal{F}[\log\varphi_R - \log\varphi_G]}_{\text{illumination}} + \underbrace{\mathcal{F}[\log\rho_R - \log\rho_G]}_{\text{reflectance}} + \underbrace{e^{i\theta_R}(a_R - a_G \cdot Cor_{RG})}_{\text{high-lit residual}}. \tag{10}$$

Leveraging on the inherent properties of the spectral separation and phase-modulated structure of residual terms, we design specialized filtering strategies that aim to robustly extract invariant illumination features, thus enhancing the reliability and effectiveness of feature extraction under varying lighting conditions.

# 4    Frequency-domain Radial Basis Network

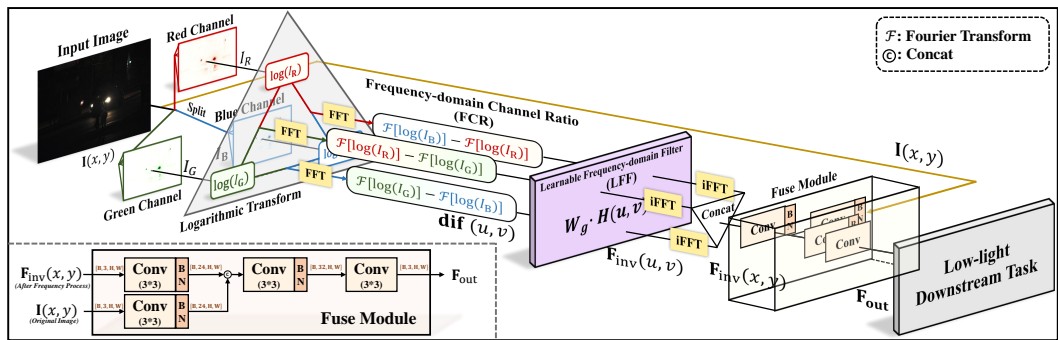

Figure 2: The overall pipeline of our proposed FRBNet. It performs illumination-invariant feature enhancement process in frequency domain using a core learnable filter for downstream low-light vision tasks.

Our theoretical analysis presented in Section 3.2 reveals that illumination interference predominantly accumulates within the low-frequency components of the signal. In contrast, the residual interference manifests as direction-dependent patterns, which are distinctly characterized by phase modulation. Thus, illumination-invariant features in real-world low-light images can be enhanced by suppressing fluctuating illumination interference and high-lit residual terms. To this end, we propose the Frequency-domain Radial Basis Network, which is a lightweight plug-and-play module as illustrated in Fig. 2. In this section, we will first introduce the whole illumination-invariant feature enhancement process based on frequency-domain channel ratio, and then provide a detailed description of the core learnable frequency-domain filter.

## 4.1    Illumination-invariant feature enhancement process in the frequency domain

To enhance illumination-invariant features, the proposed FRBNet first converts the operation of channel ratio to the frequency domain. Inter-channel relationships are exploited in the frequency domain according to the FCR function presented in Section 3.2. Define the input image in the spatial domain as $\mathbf{I}(x, y)$, for each channel pair, FCR is implemented by the frequency-domain logarithmic difference with learnable frequency parameters $(u, v)$ as:

$$\begin{cases} \text{dif}^{RG}(u, v) = \mathcal{F}[\log I_R(x, y)] - \mathcal{F}[\log I_G(x, y)] \\ \text{dif}^{GB}(u, v) = \mathcal{F}[\log I_G(x, y)] - \mathcal{F}[\log I_B(x, y)] \\ \text{dif}^{BR}(u, v) = \mathcal{F}[\log I_B(x, y)] - \mathcal{F}[\log I_R(x, y)]. \end{cases} \tag{11}$$

Next, a **L**earnable **F**requency-domain **F**ilter, defined as **LFF**, is designed to reduce the impact of illumination and high-lit residual terms in low-light images on robust feature extraction for each channel pair, which is composed of a zero-DC frequency window and an improved radial basis filter. The frequency response feature $\mathbf{F}_{\text{inv}}(u, v)$ can be expressed as:

$$\begin{cases} F_{\text{inv}}^{RG}(u, v) = LFF^{RG}(u, v) \cdot \text{dif}^{RG}(u, v) \\ F_{\text{inv}}^{GB}(u, v) = LFF^{GB}(u, v) \cdot \text{dif}^{GB}(u, v) \\ F_{\text{inv}}^{BR}(u, v) = LFF^{BR}(u, v) \cdot \text{dif}^{BR}(u, v). \end{cases} \tag{12}$$

Then, the filtered spectral features are transformed back to the spatial domain. The resulting features of all channel pairs ($R \& G$, $G \& B$, $B \& R$) are concatenated as:

$$\mathbf{F}_{\text{inv}}(x, y) = \text{Cat}\left(\mathcal{F}^{-1}\left[F_{\text{inv}}^{RG}(u, v)\right]; \mathcal{F}^{-1}\left[F_{\text{inv}}^{GB}(u, v)\right]; \mathcal{F}^{-1}\left[F_{\text{inv}}^{BR}(u, v)\right]\right), \tag{13}$$

where $\mathcal{F}^{-1}$ represents the inverse Fourier transform and $\text{Cat}$ represents the concatenation operation. To further combine the enhanced illumination invariant features from the frequency domain with the spatial-domain features from the original image, a common fuse module referring to [5] is employed for integration as:

$$\mathbf{F}_{\text{out}} = \text{Conv}\left\{\text{CB}\left[\text{Cat}\left(\text{CB}[\mathbf{F}_{\text{inv}}(x, y)]; \text{CB}[\mathbf{I}(x, y)]\right)\right]\right\}, \tag{14}$$

where Conv is a convolution while CB is a Convolution followed by a Batch Normalization (BN). Finally, the output feature $\mathbf{F}_{\text{out}}$ is fed into the downstream task network.

## 4.2 Learnable frequency-domain filter

The core of our approach is the **L**earnable **F**requency-domain **F**ilter (**LFF**) that adaptively processes spectral components. This filter consists of two complementary elements: a zero-DC frequency window $\mathbf{W_g}$ that attenuates low-frequency illumination and an improved radial basis filter $\mathbf{H}(u, v)$ that encodes both spectral distance and directional information, which can be formulated as:

$$\mathbf{LFF}(u, v) = \mathbf{W_g} \cdot \mathbf{H}(u, v). \tag{15}$$

**Zero-DC Frequency Window.** To suppress undesired illumination while preserving structural information, a Gaussian window is employed centered at the origin of the frequency plane as:

$$\mathbf{W_g}(u, v) = \exp\left(-\frac{\mathbf{r}(u, v)^2}{\sigma_w^2}\right), \quad \mathbf{r}(u, v) = \sqrt{u^2 + v^2}, \tag{16}$$

where $\sigma_w$ is a learnable bandwidth parameter, and $\mathbf{r}(u, v)$ denotes the normalized radial frequency coordinate. To eliminate the DC component, $\mathbf{W_g}(0, 0)$ is explicitly set to 0, which ensures the filter to remove global brightness offsets while retaining mid- to high-frequency information for local structural cues.

**Improved Radial Basis Filter.** To construct a spectrally adaptive and directionally selective filter, we employ a set of learnable radial basis functions (RBFs) combined with angular modulation. RBFs can capture frequency-magnitude selectivity, whereas angular terms can introduce orientation sensitivity to enable anisotropic filtering in the Fourier domain. Define a set of $K$ radial basis functions $\phi(u, v)$ centered at predefined frequency radii $\mu_k \in [0, 1]$ as:

$$\phi_k(u, v) = \exp\left(-\frac{(r(u, v) - \mu_k)^2}{2\sigma_h^2}\right), k = [1, 2, \cdots, K] \tag{17}$$

where $r(u, v)$ is the normalized radial frequency as defined earlier, and $\sigma_h$ is a learnable bandwidth parameter shared across all bases. With learnable coefficients $a_k$ of the weighted linear combination, the final radial response is:

$$\Phi(u, v) = \sum_{k=1}^{K} a_k \cdot \phi_k(u, v), k = [1, 2, \cdots, K] \tag{18}$$

Furthermore, referring to the phase-oriented residual structure in Section 3.2, the interference term exhibits dominant orientation components. The radial response is further modulated by an angular term constructed from sinusoidal harmonics of orientation angle to capture directional selectivity as:

$$M(u, v) = 1 + \lambda \cdot \sum_{n=1}^{N} \left[\cos(n\theta(u, v)) + \sin(n\theta(u, v))\right], \quad \theta(u, v) = \arctan\left(\frac{v}{u + \epsilon}\right), \tag{19}$$

where $N$ is the number of angular frequencies and $\lambda$ controls the modulation strength. The final frequency-domain radial basis filter response is given by:

$$\mathbf{H}(u, v) = \Phi(u, v) \cdot M(u, v). \tag{20}$$

By integrating angular harmonics, the improved radial basis filter is both spectrally localized and directionally responsive, enabling to alignment or suppression of such oriented residuals in a data-driven manner, which is crucial for isolating illumination-invariant features while attenuating structured interference.

## 5 Experiments

We conduct extensive experiments to evaluate the effectiveness of the proposed plug-and-play FRB-Net on low-light vision tasks of detection and segmentation. Specifically, we adopt ExDark[40],

DarkFace[73], ACDC-night[50], and LIS[4] datasets for dark object detection, face detection, night-time semantic segmentation, and dark instance segmentation tasks, respectively. Experiments are implemented based on the `MMDetection` [2] and `MMSegmentation` [6] toolboxes by PyTorch and trained on a NVIDIA RTX 4090 GPU. We select several recent representative methods for comprehensive comparison in each task. For fair comparisons, the number of radial basis functions $K$ is set to 10 and the angular modulation strength $\lambda$ is set to 0.1. Standard metrics including $\mathrm{Recall}$, $\mathrm{mAP}$, and $\mathrm{mIoU}$ are adopted for evaluation. More details for each task can be found in Appendix A.3.

## 5.1 Low-light detection tasks

**Settings.** We evaluate FRBNet on low-light detection tasks using two representative detectors: YOLOv3 [47] and TOOD [16]. Both detectors are initialized with COCO-pretrained weights and fine-tuned with FRBNet as a plug-in frontend on low-light datasets. We select representative methods from four paradigms for comparison: enhancement-based approaches, synthetic data training, multi-task learning, and plug-and-play modules. Following the experimental setup of YOLA [5], we set the momentum and weight decay of the SGD optimizer for the detection model to 0.9 and 0.0005, respectively. The learning rate is 0.001. For ExDark, all input images are resized to $608 \times 608$, and both detectors are trained for 24 epochs. For DarkFace, YOLOv3 maintains $608 \times 608$ and is trained for 20 epochs, while TOOD uses a higher resolution of $1500 \times 1000$ and is trained for 12 epochs.

**Results of Object Detection.** On the ExDark dataset, FRBNet consistently improves performance over baseline detectors and achieves the best $\mathrm{mAP}$ (see Table 1). Specifically, our method attains 90.6% $\mathrm{Recall}$ and 74.9% $\mathrm{mAP}$ with YOLOv3, surpassing the previous state-of-the-art YOLA by 0.4 $\mathrm{mAP}$. When integrated into TOOD, FRBNet further boosts performance to 93.2% $\mathrm{Recall}$ and 75.3% $\mathrm{mAP}$, outperforming all enhancement-based and multi-task approaches. For a fair comparison, part of the experimental results from YOLA [5]. These results demonstrate the effectiveness of our frequency-domain design in preserving structural cues under illumination degradation.

**Results of Dark Face Detection.** Consistent with the official experimental setup in UG2+ Challenge, we adopt a 3:1:1 random split of the DarkFace dataset for training, validation, and testing in our experiments. Table 1 presents FRBNet achieving strong performance across both detectors. It obtains 75.7% $\mathrm{Recall}$ and 57.7% $\mathrm{mAP}$ with YOLOv3, outperforming all previous plug-and-play and enhancement-based methods. For TOOD, our module improves detection performance to 82.7% $\mathrm{Recall}$ and 65.1% $\mathrm{mAP}$, setting a new state-of-the-art and exceeding the previous best (YOLA) by 2.0% $\mathrm{mAP}$. These gains highlight the generality and robustness of FRBNet across different detectors.

Table 1: Quantitative results of low-light object detection and face detection on ExDark[40] and DarkFace[73]. **Bold values** indicate the best results, while underline values represent the second-best.

| Paradigm | Method | ExDark | | | | DarkFace | | | |
| --- | --- | --- | --- | --- | --- | --- | --- | --- | --- |
| | | YOLOv3 | | TOOD | | YOLOv3 | | TOOD | |
| | | Recall | mAP | Recall | mAP | Recall | mAP | Recall | mAP |
| | Baseline | 84.6 | 71.0 | 91.9 | 72.5 | 73.8 | 54.8 | 80.9 | 57.0 |
| Enhancement | SMG[66](*CVPR-23*) | 82.3 | 68.5 | 91.8 | 71.5 | 73.4 | 52.4 | 80.2 | 56.3 |
| | NeRCo[70](*ICCV-23*) | 83.4 | 68.5 | 91.8 | 71.8 | 73.8 | 53.0 | 79.4 | 56.8 |
| | LightDiff[25](*ECCV-24*) | 84.3 | 71.3 | 92.1 | 72.9 | 75.5 | 57.4 | 81.0 | 58.7 |
| | DarkIR[15](*CVPR-25*) | 81.9 | 68.2 | 90.9 | 72.0 | 74.5 | 55.9 | 81.4 | 60.4 |
| Synthetic Data | DAINet*[13](*CVPR-24*) | 86.7 | 73.4 | - | - | 74.8 | 56.9 | - | - |
| | WARLearn[1](*WACV-25*) | 85.6 | 72.4 | 92.8 | 73.4 | 74.5 | 56.2 | 80.8 | 59.4 |
| Multi-task | MAET[10](*ICCV-21*) | 85.1 | 72.5 | 92.5 | 74.3 | 74.7 | 55.7 | 80.7 | 59.6 |
| | IAT[9](*BMVC-22*) | 85.0 | 72.6 | 92.9 | 73.0 | 73.6 | 55.5 | 79.7 | 58.3 |
| Plug-and-play | DENet[46](*ACCV-22*) | 84.2 | 71.3 | 92.6 | 73.5 | 71.8 | 52.6 | 73.6 | 49.6 |
| | FeatEnHancer[21](*ICCV-23*) | 90.4 | 71.2 | **96.4** | 74.6 | 74.1 | 55.2 | 81.7 | 60.5 |
| | YOLA[5](*NeurIPS-24*) | 86.1 | 72.7 | 93.8 | 75.2 | 74.9 | 56.3 | **83.1** | 63.2 |
| | **FRBNet(ours)** | **90.6** | **74.9** | 93.2 | **75.4** | **75.7** | **57.7** | 82.7 | **65.1** |

## 5.2 Low-light segmentation tasks

**Settings.** We assess the ability of FRBNet to perform low-light segmentation tasks in nighttime semantic segmentation and dark instance segmentation. For the semantic segmentation task, the input images of ACDC-Night [50] are resized to $2048 \times 1024$. DeepLabV3+ [3] is adopted as the baseline

with a ResNet-50 [55] backbone, which is initialized with ImageNet-pretrained weights [11]. We compare our FRBNet with current state-of-the-art methods. Following the experimental protocol of FeatEnHancer [21], all methods are trained for 20K iterations. The comparison includes traditional enhancement-based methods as well as more recent task-oriented approaches. For the instance segmentation task, the input images of the LIS dataset [4] are resized to $1330 \times 800$. Mask R-CNN [22] with a ResNet-50 backbone implemented via the `MMDetection` framework [2] is employed as the baseline model. MBLLEN [41], DarkIR [15], FeatEnHancer [21], and YOLA [5] are selected as comparative models. All models are trained for 24 epochs using the SGD optimizer.

Table 2: Quantitative results of low-light semantic segmentation on the ACDC[50]. The symbol set {RO,SI,BU,WA,FE,PO,TL,ST,VE,TE,SK,PE,CA,TR,BI} represents {*road*, *sidewalk*, *building*, *wall*, *fence*, *pole*, *traffic light*, *traffic sign*, *vegetation*, *terrain*, *sky*, *person*, *car*, *train*, *bicycle*}.

| Method | RO | SI | BU | WA | FE | PO | TL | TS | VE | TE | SK | PE | CA | TR | BI | mIoU |
|---|---|---|---|---|---|---|---|---|---|---|---|---|---|---|---|---|
| Baseline[3] | 90.0 | 61.4 | 74.2 | 32.8 | 34.4 | 45.7 | 49.8 | 31.2 | 68.8 | 14.6 | 80.4 | 27.1 | 62.1 | 76.3 | 14.4 | 50.8 |
| RetinexNet[63] | 89.4 | 61.0 | 70.6 | 30.1 | 28.1 | 42.4 | 47.6 | 25.7 | 65.8 | 8.6 | 77.3 | 21.5 | 54.8 | 67.4 | 8.2 | 46.5 |
| DRBN[65] | 90.5 | 61.5 | 72.8 | 31.9 | 32.5 | 44.5 | 47.3 | 27.2 | 65.7 | 10.2 | 76.5 | 24.2 | 55.4 | 71.1 | 11.9 | 48.2 |
| FIDE[71] | 90.0 | 60.7 | 72.8 | 32.4 | 34.1 | 43.3 | 47.9 | 26.1 | 67.0 | 13.7 | 78.0 | 26.5 | 57.1 | 71.0 | 12.4 | 48.8 |
| KinD[75] | 90.0 | 61.0 | 73.2 | 31.9 | 32.8 | 43.5 | 42.7 | 27.7 | 65.5 | 13.3 | 77.4 | 22.8 | 55.1 | 74.5 | 11.5 | 48.1 |
| EnGAN[26] | 89.7 | 58.9 | 73.7 | 32.8 | 31.8 | 44.7 | 49.2 | 26.2 | 67.3 | 14.2 | 77.8 | 25.0 | 59.0 | 71.2 | 7.8 | 48.6 |
| ZeroDCE[20] | 90.6 | 59.9 | 73.9 | 32.6 | 31.7 | 44.3 | 46.2 | 25.8 | 67.2 | 14.6 | 79.1 | 24.7 | 59.4 | 66.8 | 13.9 | 48.7 |
| SSIENet[76] | 89.6 | 59.3 | 72.5 | 29.9 | 31.7 | 45.4 | 43.9 | 24.5 | 66.7 | 10.6 | 78.3 | 22.8 | 52.6 | 71.1 | 5.4 | 46.9 |
| Xue *et al.*[68] | 93.2 | 72.6 | 78.4 | 43.8 | 46.5 | 48.1 | 51.1 | 38.8 | 68.6 | 14.9 | 79.1 | 21.9 | 61.6 | 85.2 | 36.1 | 55.8 |
| FeatEnHancer[21] | 93.5 | 70.6 | 75.6 | 41.8 | 33.4 | 51.3 | 55.2 | 35.9 | 68.5 | 13.4 | 80.6 | 27.6 | 61.8 | 80.0 | 51.2 | 56.0 |
| YOLA[5] | 93.2 | 72.1 | 79.3 | 41.1 | 39.1 | 53.1 | 60.4 | 44.4 | 71.5 | 4.7 | 83.2 | 37.8 | 66.8 | 85.0 | 49.2 | 58.7 |
| **FRBNet(ours)** | **94.4** | **75.5** | **79.7** | **46.0** | 45.4 | 52.3 | **64.9** | 50.8 | **72.2** | 9.5 | **84.2** | 40.9 | 70.4 | 88.7 | 49.3 | **61.6** |

**Results of Semantic Segmentation.** Table 2 summarizes the quantitative results on the ACDC-Night benchmark. Since the testing set of ACDC-Night contains some extremely rare samples, we report the quantitative results of IoU on 15 categories, excluding *truck*, *bus*, *rider*, and *motorcycle*. And the results of mIoU are adopted directly from the output of `MMSegmentation` toolbox. From Table 2, most of the existing enhancement methods yield only marginal improvements. Compared with YOLA (58.7%), FRBNet further improves to 61.6% mIoU, achieving the best result. Notably, FRBNet delivers consistent gains across multiple key classes in nighttime semantic segmentation, such as *sidewalk* (75.5%), *building* (79.7%), and *traffic sign* (50.8%). The second line in Fig. 3(a) also reveals that the visualized results of FRBNet are the most similar to the ground truth.

**Results of Instance Segmentation.** Following common practice, we evaluate instance segmentation performance with mAP, $\text{mAP}_{50}$, and $\text{mAP}_{75}$ metrics. As shown in Table 3, FRBNet achieves the best performance across all metrics on the LIS. It obtains 30.2% mAP, 50.5% $\text{mAP}_{50}$, and 30.4% $\text{mAP}_{75}$, outperforming previous methods by a clear margin.

Table 3: Quantitative results of low-light instance segmentation on the LIS[4].

| Method | mAP | $\text{mAP}_{50}$ | $\text{mAP}_{75}$ |
|---|---|---|---|
| Mask RCNN[22] | 23.7 | 41.5 | 23.3 |
| MBLLEN[41] | 22.5 | 40.7 | 22.3 |
| DarkIR[15] | 27.4 | 46.3 | 27.5 |
| YOLA[5] | 24.9 | 44.8 | 24.2 |
| FeatEnHancer[21] | 29.1 | 48.7 | 29.7 |
| **FRBNet(ours)** | **30.2** | **50.5** | **30.4** |

Table 4: Ablation study on the effectiveness of each component in FRBNet.

| | $\mathbf{H}(u,v)$ | $\mathbf{W_g}$ | FCR | ExDark | DarkFace |
|---|---|---|---|---|---|
| Baseline | | | | 71.0 | 57.0 |
| Ablation Cases | ✓ | | | 72.5 | 62.0 |
| | ✓ | ✓ | | 72.9 | 62.5 |
| | ✓ | | ✓ | 73.5 | 63.7 |
| **FRBNet** | ✓ | ✓ | ✓ | **74.9** | **65.1** |

## 5.3 Ablation studies

**Effectiveness of each component.** We evaluate each component of FRBNet with YOLOv3 on ExDark and TOOD on DarkFace. Specifically, the channel operation in the frequency domain (FCR) and the two elements of **LFF** are evaluated. As in Table 4, the proposed FRBNet adopting the whole LFF and FCR presents superior performance, and FCR plays a more relatively important role.

**Efficiency-Performance analysis.** Table 5 compares FRBNet with existing methods for the effective balance of computational efficiency and performance in low-light vision applications. Non-

architectural methods, which enhance performance through preprocessing or pretraining without modifying the detector structure, show limited performance despite their high computational efficiency. For end-to-end modules, FRBNet achieves the highest detection performance (74.9 mAP on ExDark using YOLOv3) and segmentation accuracy (61.6 mIoU) at a relatively low computational cost. FRBNet also demonstrates strong inference speed (89.5 FPS), significantly faster than others like FeatEnHancer (33.1 FPS), while achieving 3.7 mAP and 5.6 mIoU improvements.

Table 5: Efficiency-Performance trade-off of different low-light vision methods.

| Category | Metric | Non-architectural Methods | | | | End-to-End Trained Plug-and-Play Module | | | |
|---|---|---|---|---|---|---|---|---|---|
| | | KinD[75] | Zero-DCE[20] | SMG[66] | MAET[10] | DENet[46] | FeatEnHancer[21] | YOLA[5] | FRBNet |
| Efficiency | # Params ↓ | 8.2M | 79K | 17.9M | 40M | 40K | 138K | **8K** | 9K |
| | Flops(G) ↓ | 50.6 | | | | 61.7 | 79.5 | 55.0 | 53.1 |
| | FPS(img/s)↑ | 95.8 | | | | 83.8 | 33.1 | 81.1 | 89.5 |
| Performance | Det(mAP) ↑ | 69.4 | 71.1 | 68.5 | 72.5 | 71.3 | 71.2 | 72.7 | **74.9** |
| | Seg(mIoU) ↑ | 48.1 | 48.7 | 49.7 | - | 52.2 | 56.0 | 58.7 | **61.6** |

## 5.4 Visualization

The visualization of experiment results and feature maps on ExDark are presented in Fig. 3. The top line in Fig. 3(a) verifies FRBNet achieves the most accurate detection. In Fig. 3(b), FeatEnHancer brings color deviation artifacts, and YOLA struggles with fine details in low-light regions. FRBNet generates more balanced feature representations with better preservation of object boundaries and structural details, especially in the outlines. From the heatmaps in Fig. 3(c), compared to the Baseline, our method produces more spatially focused feature responses, particularly around object contours such as the bicycle frame and the human head, which allows FRBNet to preserve fine object details and reveal richer gradient variations. Our approach successfully isolates illumination-invariant features, thereby enhancing robustness for downstream tasks.

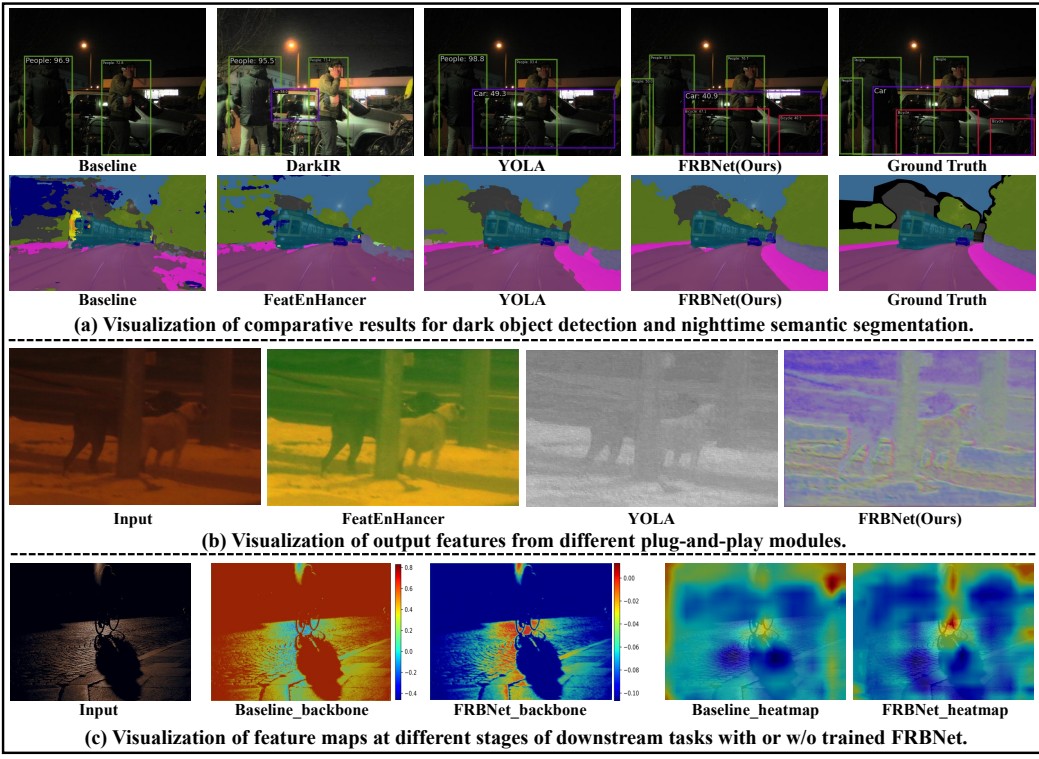

Figure 3: Qualitative results. (a) Visualization of comparative results for dark object detection on ExDark (top) and nighttime semantic segmentation on ACDC-Night (bottom). (b)Visualization of output features from different plug-and-play modules. (c) Visualization of feature maps at different stages of downstream tasks with or without trained FRBNet.

# 6  Conclusion

This paper presents FRBNet, a novel frequency-domain framework for extracting illumination-invariant features in low-light conditions by leveraging learnable radial basis filters with frequency-channel operations. This plug-and-play module can be seamlessly integrated into existing architectures and achieves significant performance improvements. Based on extensive experimental demonstrations, FRBNet can effectively address the limitations of spatial-domain approaches for low-light downstream tasks. Future research will focus on optimizing the universality of modules and exploring broader application scenarios to further advance the development of low-light vision.

## Acknowledgments and Disclosure of Funding

We would like to thank the anonymous reviewers for their valuable suggestions and comments. This work was supported by the National Natural Science Foundation of China (NSFC) under grant Nos.62206307 and 12401590.

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

# A Technical Appendices and Supplementary Material

## A.1 Revisiting Imaging Principle of Low-light Vision

Our extension of the Lambertian model draws significant inspiration from the Phong model's additive component approach. The Phong lighting model [45] provides a comprehensive framework for simulating light-surface interactions through additive component decomposition. This model serves as the theoretical foundation for our extended Lambertian formulation in the main text, particularly in our treatment of non-uniform highlights in low-light imagery.

The cornerstone of the Phong model is its decomposition of surface illumination into three distinct additive components:

$$I = I_a + I_d + I_s, \tag{21}$$

where, $I_a$ represents ambient reflection component, $I_d$ represents diffuse reflection component, and $I_s$ represents specular reflection component.

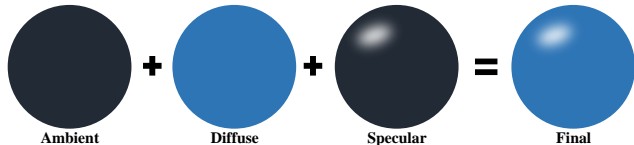

Figure 4: Illustration of Phong Lighting Model Imaging Mechanism

This additive decomposition directly inspired our approach in Eq. 2 of the main text, where we extended the traditional Lambertian model by adding a spatially irregular highlight component $S_C(x, y)$.

The standard Phong model was originally developed for controlled lighting environments in computer graphics rendering. Our approach extends this concept to address the unique challenges of real-world low-light imagery. While the Phong model assumes idealized light sources and surface properties, actual low-light scenes contain complex lighting elements like streetlights, vehicle headlights, and neon signs that create irregular highlight patterns.

To account for these complexities, we adapt the Phong model's additive principle by introducing a spatially varying highlight term $H_C(x, y)$ that modulates the base diffuse reflection. This modification preserves the fundamental additive relationship between components while providing the flexibility needed to represent the non-uniform highlight distributions characteristic of natural low-light environments. By building upon the Phong model's decomposition approach rather than strictly adhering to its original formulation, our extended Lambertian model can more accurately represent the complex illumination patterns found in real-world low-light imagery.

Figure 4 illustrates the Phong model's imaging mechanism, visually demonstrating how the three components (ambient, diffuse, and specular) combine additively to create the final rendered image. This decomposition serves as the theoretical foundation for our treatment of complex lighting in low-light scenes described in the main text.

## A.2 Derivation of Frequency Correlation Coefficient

The characterization of inter-channel relationships in the frequency domain is crucial for understanding how highlight interference manifests across different color channels. While our previous analysis identified the presence of phase-modulated residual components, we need a precise mathematical formulation to quantify this directional phenomenon. To this end, we introduce the frequency correlation coefficient $Cor_{RG}$ that captures the angular displacement between channel responses. The following derivation formalizes this relationship in the frequency domain, providing the mathematical basis for our angular-modulated filtering approach. Given an image $I_C(x, y)$, where $C \in R, G, B$ denotes the color channel, we define its two-dimensional Discrete Fourier Transform (DFT) as:

$$\mathcal{I}_C(u, v) = \mathcal{F}[I_C(x, y)] = \frac{1}{wh} \sum_{x=0}^{w-1} \sum_{y=0}^{h-1} I_C(x, y) e^{-i2\pi(\frac{ux}{w} + \frac{vy}{h})}, \tag{22}$$

where $h$ and $w$ denote the height and width of the image, and $\mathcal{I}_C(u, v)$ represents the complex Fourier coefficient at frequency component $(u, v)$. Each Fourier coefficient encodes both magnitude and phase information, which can be separated using the complex exponential representation:

$$\mathcal{I}_C(u, v) = |\mathcal{I}_C(u, v)|e^{i\phi_C(u,v)}, \tag{23}$$

where $|\mathcal{I}_C(u, v)|$ denotes the magnitude and $\phi_C(u, v)$ represents the phase at frequency $(u, v)$. To further characterize the structural consistency between color channels in the frequency domain, we follow the derivation of complex correlation coefficients based on amplitude and phase analysis [56]. For clarity, let us denote the magnitude as $\alpha_C = |\mathcal{I}_C(u, v)|$ and the phase as $\rho_C = \phi_C(u, v)$ for each channel $C$. We can then express the Fourier coefficient as:

$$\mathcal{I}_C(u, v) = \alpha_C \cos(\rho_C) + i\alpha_C \sin(\rho_C) = \alpha_C \cdot e^{i\rho_C}. \tag{24}$$

The magnitude of this complex coefficient can be verified as:

$$|\mathcal{I}_C(u, v)| = \sqrt{\alpha_C^2(\cos^2(\rho_C) + \sin^2(\rho_C))} = \alpha_C \tag{25}$$

For two frequency responses at the same spatial frequency $(u, v)$ from distinct channels, e.g., $\mathcal{I}_R(u, v) = \alpha_R e^{i\rho_R}$ and $\mathcal{I}_G(u, v) = \alpha_G e^{i\rho_G}$, we define the complex correlation coefficient as:

$$Cor_{RG}(u, v) = \frac{\mathcal{I}_R(u, v) \cdot \mathcal{I}_G(u, v)}{|\mathcal{I}_R(u, v)| \cdot |\mathcal{I}_G(u, v)|}, \tag{26}$$

where $\mathcal{I}_G(u, v)$ is the complex conjugate of $\mathcal{I}_G(u, v)$, computed as $\mathcal{I}_G^*(u, v) = \alpha_G e^{-i\rho_G}$. Expanding this equation:

$$Cor_{RG}(u, v) = \frac{\alpha_R e^{i\rho_R} \cdot \alpha_G e^{-i\rho_G}}{\alpha_R \cdot \alpha_G} \tag{27}$$

$$= \frac{\alpha_R \alpha_G e^{i(\rho_R - \rho_G)}}{\alpha_R \alpha_G} \tag{28}$$

$$= e^{i(\rho_R - \rho_G)} \tag{29}$$

$$= e^{i\Delta\rho} \tag{30}$$

where $\Delta\rho = \rho_R - \rho_G$ represents the phase difference between the R and G channels at frequency $(u, v)$. This elegant formulation reveals a fundamental insight: the correlation between channels at each frequency location is directly encoded by their phase difference $\Delta\rho$. The correlation coefficient $Cor_{RG}(u, v)$ has unit magnitude but carries critical directional information:

- When $\Delta\rho = 0$, $Cor_{RG}(u, v) = 1$, it indicates perfect phase alignment between channels.
- When $\Delta\rho = \pi$, $Cor_{RG}(u, v) = -1$, it reveals exactly opposite phases.
- When $\Delta\rho = \pm\frac{\pi}{2}$, $Cor_{RG}(u, v) = \pm i$, it corresponds to orthogonal phase relationships.

The correlation coefficient can also be expressed in terms of its real and imaginary components:

$$Cor_{RG}(u, v) = \cos(\Delta\rho) + i\sin(\Delta\rho). \tag{31}$$

This phase-based correlation measure provides crucial insights into the directional patterns of highlight interference across color channels. In our extended Lambertian model with highlight interference, these phase differences encode the angular displacement of interference patterns, which cannot be captured by simple magnitude-based analysis. By incorporating this correlation coefficient into our frequency-domain filter design, we enable directionally-aware processing that adapts to the specific phase relationships induced by complex lighting conditions. This theoretical foundation directly informs our angular-modulated filtering approach, allowing us to effectively isolate illumination-invariant features even in the presence of highly directional highlight interference.

### A.3 Implementation Details

**Statistics of the Datasets** Table 6 summarizes the statistics of our employed datasets. These datasets cover a wide range of low-light vision tasks, including object detection, face detection, semantic segmentation, and instance segmentation. ExDark[40] is one of the most widely used benchmarks for

dark object detection, featuring diverse scenes and object categories under extremely low-light conditions. Dark Face[73] focuses on the challenging task of face detection in dark environments, providing densely annotated facial regions. ACDC-Night[50] targets nighttime semantic segmentation, with a particular emphasis on road scenes, making it valuable for autonomous driving applications. LIS[4] is a recently proposed dataset designed for low-light instance segmentation, offering fine-grained annotations in real-world dark scenarios. The combination of these datasets enables a comprehensive evaluation across different low-light vision tasks. # Class is the number of classes, whereas #Train and #Val denote the number of training and validation samples for each dataset, respectively.

Table 6: Statistics of the datasets

| Dataset | Task | # Class | # Train | # Val |
|---------|------|---------|---------|-------|
| ExDark[40] | Dark object detection | 12 | 3000 | 1800 |
| Dark Face[73] | Dark face detection | 1 | 3600 | 1200 |
| ADCD-Night[50] | Nighttime semantic segmentation | 15 | 400 | 106 |
| LIS[4] | Low-light instance segmentation | 8 | 1561 | 669 |

**FRBNet on Dark Object Detection.** For all experiments, we adopt the official implementations of YOLOv3 and TOOD detectors with standardized training protocols. The YOLOv3 detector uses a Darknet-53 backbone pre-trained on ImageNet, while TOOD employs a ResNet-50 backbone with FPN. Both models are trained for 24 epochs using the SGD optimizer with momentum 0.9 and weight decay 5e-4. The learning rate begins at 0.001 with a linear warm-up for the first 1000 iterations. We apply standard data augmentation techniques, including random expansion, minimum IoU random cropping, random resizing, random flipping, and photometric distortion. For testing, images are resized to 608×608 maintaining the aspect ratio. We use a batch size of 8 on a single GPU.

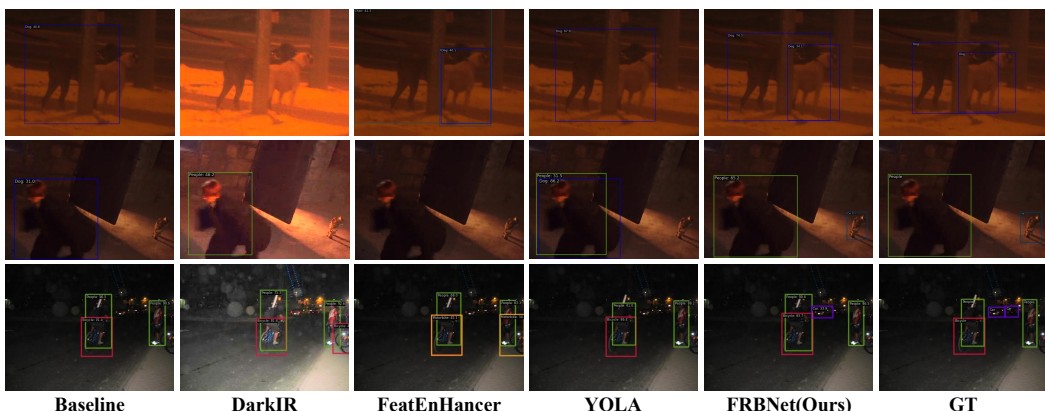

| Baseline | DarkIR | FeatEnHancer | YOLA | FRBNet(Ours) | GT |

Figure 5: Qualitative comparisons of dark object detection methods on ExDark dataset.

Table 5 and 6 present comprehensive quantitative comparisons on the ExDark dataset using YOLOv3 and TOOD detectors, respectively. We evaluate detection performance across all 12 object categories, reporting both category-specific Average Precision (AP) and overall mean Average Precision ($mAP_{50}$). FRBNet demonstrates particularly strong improvements on challenging categories such as "Bottle" (+2.6% over YOLA with YOLOv3), "Bus" (+1.8% over DAINet with YOLOv3), and "Chair" (+2.3% over DAINet with YOLOv3). These categories typically involve smaller objects or objects with challenging contrast profiles in low-light conditions, suggesting that our frequency-domain processing effectively preserves discriminative features for these difficult cases. Figure 5 presents qualitative comparisons of detection results from various methods on challenging ExDark samples. As shown in the visualization, our FRBNet achieves more accurate object localization and higher detection confidence compared to the baseline, DarkIR, FeatEnHancer, and YOLA methods.

**FRBNet on Dark Face Detection.** We continue with YOLOv3 and TOOD as base detectors, largely following the experimental setup from YOLA[5]. Since the UG2+ Challenge concluded in 2024, we adopted a standard random split with a 3:1:1 ratio for training, validation, and testing. Our implementation is based on the `MMDetection` framework with customized data pipelines. For

Table 7: Quantitative comparisons of the ExDark[40] dataset based on YOLOv3 detector.

| Method | Bicycle | Boat | Bottle | Bus | Car | Cat | Chair | Cup | Dog | Motorbike | People | Table | mAP$_{50}$ |
|---|---|---|---|---|---|---|---|---|---|---|---|---|---|
| Baseline [47] | 79.8 | 72.1 | 70.9 | 82.8 | 79.5 | 64.4 | 67.6 | 70.6 | 79.5 | 62.4 | 77.7 | 44.2 | 71.0 |
| MBLLEN [41] | 77.5 | 72.5 | 70.2 | 80.7 | 80.6 | 65.0 | 65.2 | 70.6 | 77.9 | 64.9 | 77.3 | 41.8 | 70.3 |
| KIND [75] | 80.2 | 74.4 | 71.5 | 81.0 | 80.3 | 62.2 | 61.3 | 67.5 | 75.8 | 62.1 | 75.9 | 40.9 | 69.4 |
| Zero-DCE [20] | 81.8 | 74.6 | 70.1 | 86.3 | 79.5 | 61.0 | 66.2 | 71.7 | 78.4 | 62.9 | 77.3 | 43.1 | 71.1 |
| EnlightenGAN [26] | 81.1 | 74.2 | 69.8 | 83.3 | 78.3 | 63.3 | 65.5 | 69.3 | 75.3 | 62.5 | 76.7 | 41.0 | 70.0 |
| RUAS [36] | 76.4 | 69.2 | 62.7 | 77.3 | 74.9 | 59.0 | 64.3 | 64.8 | 73.1 | 55.8 | 71.5 | 38.8 | 65.7 |
| SCI [43] | 80.3 | 74.2 | 73.6 | 82.8 | 78.4 | 64.4 | 65.8 | 71.3 | 78.1 | 62.7 | 78.2 | 42.4 | 71.0 |
| NeRCo [70] | 80.8 | 73.6 | 66.3 | 81.3 | 75.6 | 62.8 | 62.5 | 67.7 | 75.6 | 61.8 | 75.1 | 39.0 | 68.5 |
| SMG [66] | 78.1 | 72.1 | 65.8 | 81.6 | 78.3 | 63.7 | 64.5 | 67.6 | 76.3 | 57.4 | 73.7 | 42.4 | 68.5 |
| LightDiff [25] | 81.7 | 74.1 | 73.3 | 85.2 | 80.2 | 62.5 | 67.3 | 71.4 | 74.7 | 63.5 | 75.8 | 46.1 | 71.3 |
| DarkIR [15] | 78.5 | 73.3 | 66.0 | 84.9 | 76.8 | 59.4 | 62.9 | 65.1 | 74.3 | 62.0 | 73.7 | 41.9 | 68.2 |
| DENet [46] | 81.1 | 75.0 | 73.9 | 87.1 | 79.7 | 63.5 | 66.3 | 69.6 | 76.3 | 61.4 | 76.7 | 44.9 | 71.3 |
| PENet [74] | 76.5 | 71.9 | 67.4 | 84.2 | 78.0 | 59.9 | 64.6 | 66.7 | 74.8 | 62.5 | 73.9 | 45.1 | 68.8 |
| MAET [10] | 81.5 | 73.7 | 74.0 | 88.2 | 80.9 | 68.8 | 66.9 | 71.8 | 79.3 | 60.2 | 78.8 | 46.3 | 72.5 |
| FeatEnHancer[21] | 79.7 | 75.9 | 73.3 | 87.5 | 81.2 | 62.0 | 64.9 | 67.9 | 75.7 | 64.2 | 76.6 | 45.3 | 71.2 |
| DAINet [13] | 81.1 | 77.7 | 74.1 | 89.4 | 80.4 | 68.6 | 69.3 | 71.1 | 81.5 | 65.3 | 78.6 | 45.1 | 73.5 |
| YOLA [5] | 82.4 | 74.0 | 72.7 | 85.4 | 81.0 | 67.2 | 66.5 | 71.5 | 81.8 | 65.2 | 78.6 | 45.7 | 72.7 |
| FRBNet(our) | 84.3 | 75.6 | 75.3 | 89.8 | 82.0 | 68.6 | 71.6 | 74.8 | 82.6 | 65.8 | 81.0 | 46.5 | 74.9 |

Table 8: Quantitative comparisons of the ExDark[40] dataset based on TOOD detector.

| Method | Bicycle | Boat | Bottle | Bus | Car | Cat | Chair | Cup | Dog | Motorbike | People | Table | mAP$_{50}$ |
|---|---|---|---|---|---|---|---|---|---|---|---|---|---|
| Baseline [16] | 80.6 | 75.8 | 71.1 | 88.1 | 76.8 | 70.4 | 66.8 | 69.2 | 85.4 | 61.5 | 76.1 | 48.2 | 72.5 |
| MBLLEN [41] | 80.8 | 77.8 | 72.8 | 89.3 | 78.7 | 73.5 | 67.5 | 69.4 | 85.2 | 62.9 | 77.3 | 47.2 | 73.5 |
| KIND [75] | 81.7 | 77.7 | 70.3 | 88.4 | 78.1 | 69.7 | 67.2 | 67.8 | 84.1 | 61.6 | 76.6 | 47.8 | 72.6 |
| Zero-DCE [20] | 81.8 | 79.0 | 72.9 | 89.6 | 77.9 | 71.9 | 68.5 | 69.8 | 84.8 | 62.9 | 78.0 | 49.5 | 73.9 |
| EnlightenGAN [26] | 80.7 | 77.6 | 70.4 | 88.8 | 76.9 | 70.6 | 67.9 | 68.7 | 84.4 | 62.2 | 77.5 | 49.6 | 73.0 |
| RUAS [36] | 78.4 | 74.3 | 67.4 | 85.1 | 72.4 | 67.7 | 67.3 | 65.2 | 77.9 | 56.1 | 73.4 | 47.0 | 69.4 |
| SCI [43] | 81.3 | 78.1 | 71.6 | 89.4 | 77.6 | 71.1 | 68.0 | 70.9 | 85.0 | 63.0 | 77.2 | 49.2 | 73.5 |
| NeRCo [70] | 78.8 | 75.6 | 70.8 | 87.6 | 75.7 | 69.1 | 66.8 | 69.5 | 82.5 | 59.9 | 76.0 | 49.3 | 71.8 |
| SMG [66] | 78.2 | 75.9 | 69.9 | 87.3 | 75.1 | 71.3 | 66.5 | 67.2 | 84.2 | 60.1 | 75.1 | 46.7 | 71.5 |
| LightDiff [25] | 81.1 | 77.8 | 74.4 | 89.5 | 79.2 | 72.0 | 67.6 | 70.9 | 86.1 | 62.5 | 77.2 | 49.0 | 72.9 |
| DarkIR [15] | 78.4 | 78.0 | 70.4 | 88.7 | 76.0 | 70.8 | 67.9 | 66.5 | 83.7 | 59.5 | 75.2 | 49.0 | 72.0 |
| DENet [46] | 80.9 | 78.2 | 70.9 | 88.3 | 77.5 | 71.6 | 67.2 | 70.3 | 87.3 | 62.0 | 77.3 | 49.9 | 73.5 |
| PENet [74] | 76.0 | 72.3 | 66.7 | 84.4 | 72.2 | 65.4 | 63.3 | 65.8 | 79.1 | 53.1 | 71.0 | 44.6 | 67.8 |
| MAET [10] | 80.5 | 77.3 | 74.0 | 90.1 | 78.3 | 73.4 | 69.6 | 70.7 | 86.6 | 64.4 | 77.6 | 48.5 | 74.3 |
| FeatEnHancer[21] | 83.6 | 77.4 | 74.8 | 89.6 | 79.3 | 72.6 | 68.2 | 72.5 | 85.5 | 63.8 | 78.0 | 49.6 | 74.6 |
| YOLA [5] | 83.9 | 78.7 | 75.3 | 88.8 | 79.0 | 73.4 | 69.9 | 71.9 | 86.8 | 66.3 | 78.3 | 49.8 | 75.2 |
| FRBNet(our) | 83.2 | 78.9 | 76.5 | 91.2 | 80.7 | 74.1 | 69.9 | 72.6 | 84.6 | 64.6 | 78.6 | 49.8 | 75.4 |

training, we apply data augmentation including random expansion (ratio range 1-2), minimum IoU random cropping (IoU thresholds from 0.4 to 0.9), random resizing between (750×500) and (1500×1000) with preserved aspect ratio, and random horizontal flipping with 0.5 probability. During testing, images are resized to 1500×1000 while maintaining aspect ratio. We train YOLOv3 for 20 epochs and TOOD for 12 epochs using the SGD optimizer. Since DarkFace contains only face annotations, we configure the detectors for single-class detection.

Figure 6 presents qualitative comparisons of face detection results on challenging DarkFace samples. As shown, enhancement-based methods like LightDiffusion and DarkIR improve image visibility but often introduce artifacts or over-enhancement that can lead to false positives. FeatEnHancer, YOLA, and our FRBNet all maintain the original low-light appearance while accurately detecting faces. Notably, our method achieves more precise bounding box localization and higher detection confidence scores, particularly for faces in extremely dark regions.

**FRBNet on Nighttime Semantic Segmentation.** We further evaluate our approach on nighttime semantic segmentation using the ACDC-Night dataset to demonstrate the versatility of FRBNet across different low-light vision tasks. For semantic segmentation experiments, we adopt the MMSegmentation framework with DeepLabV3+ architecture, employing a ResNet-50 backbone initialized with ImageNet pre-trained weights. This configuration allows for direct comparison with previous state-of-the-art methods on nighttime segmentation. During training, images are resized to 2048×1024

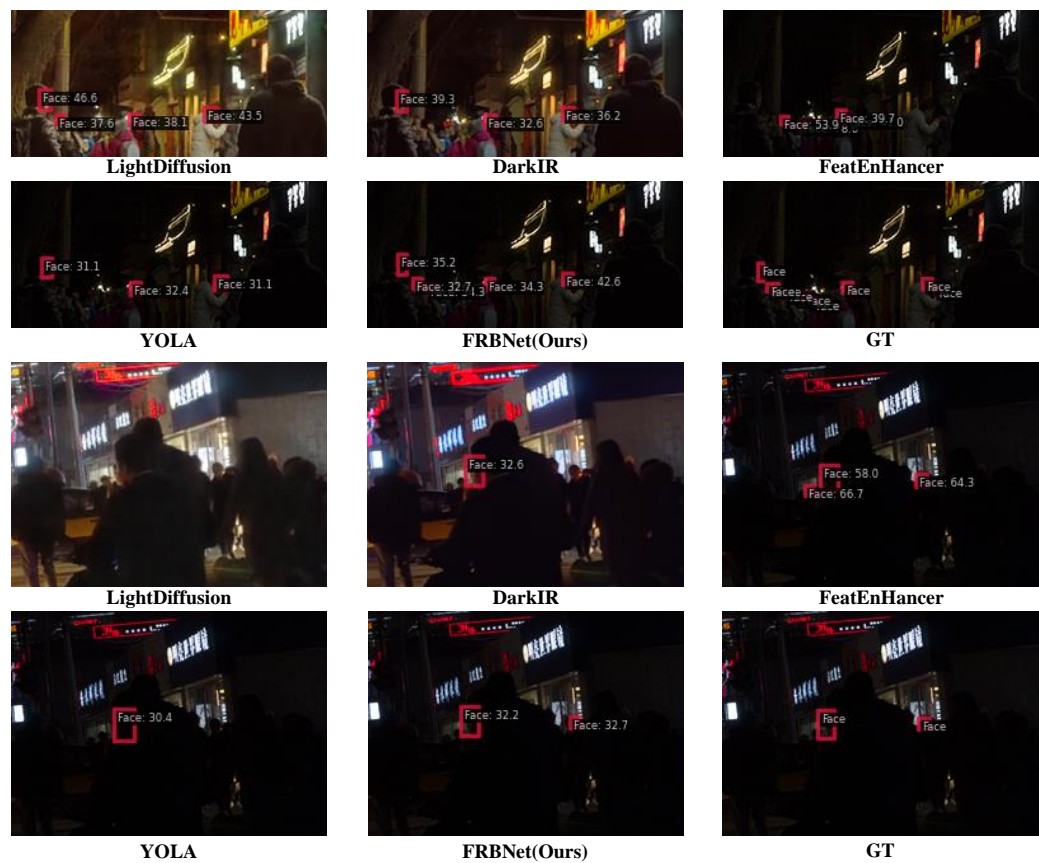

Figure 6: Qualitative comparisons of dark face detection methods on DarkFace dataset.

resolution, and we use a batch size of 4. The network is optimized using SGD with a base learning rate of 0.01 and weight decay of 0.0005, following a 20K iteration training schedule.

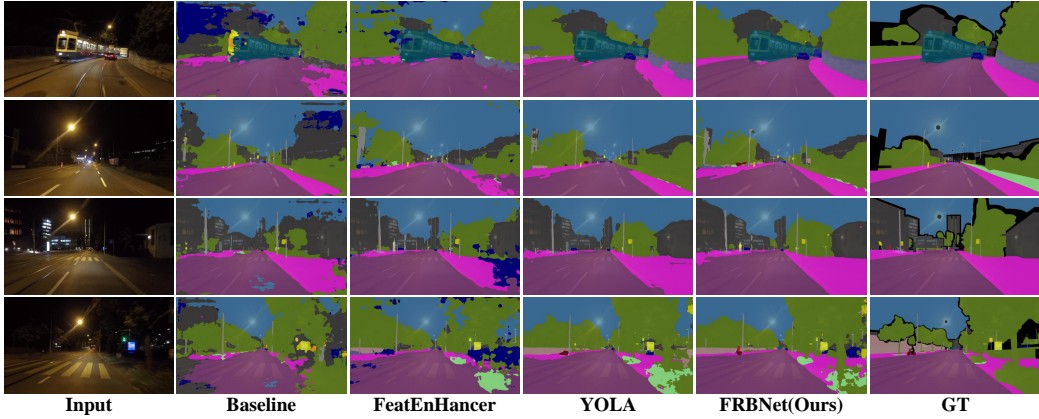

Figure 7: Qualitative comparisons of semantic segmentation methods on ACDC-Night dataset.

Figure 7 presents qualitative comparisons of segmentation results on the ACDC-Night dataset. The visualization reveals significant differences in segmentation quality across methods. The baseline method struggles with class boundaries in low-light conditions, producing fragmented and inconsistent segments, particularly visible in the second and third rows. FeatEnHancer improves overall

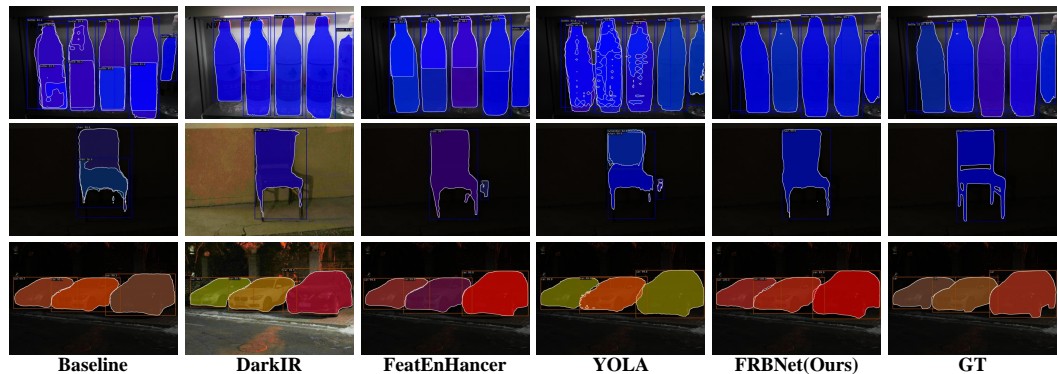

Figure 8: Qualitative comparisons of instance segmentation methods on LIS dataset.

segmentation but still misclassifies certain regions, especially in areas with strong light sources or deep shadows. YOLA produces more coherent results but exhibits some boundary inaccuracies and class confusion in complex scenes. In contrast, our FRBNet generates segmentation maps that more closely align with ground truth, maintaining consistent class boundaries even in extremely dark regions. This is especially evident in challenging scenarios like road boundaries under street lighting, distant buildings in minimal ambient light, and complex urban scenes with mixed lighting sources.

**FRBNet on Low-light Instance Segmentation.** To further evaluate the versatility of our approach across a wider range of low-light vision tasks, we conduct experiments on the Low-light Instance Segmentation dataset, which requires both object detection and instance-level segmentation in challenging illumination conditions. We employ Mask R-CNN with ResNet-50 backbone as our base architecture, implemented using the `MMDetection` framework. During training, images are resized to 1333×800 while maintaining aspect ratio, and standard random horizontal flipping is applied with a probability of 0.5. We train all models with a batch size of 8 using SGD optimizer with an initial learning rate of 0.01, momentum of 0.9, and weight decay of 0.0001. The learning rate schedule consists of a linear warm-up phase for the first 1000 iterations, followed by a multi-step decay, reducing the learning rate by a factor of 0.1 at epoch 18. All models are trained for 24 epochs with mixed precision training enabled.

Table 9: Quantitative comparisons of the Low-light Instance Segmentation[4] dataset based on Mask RCNN.

| Method | $mAP^{seg}$ | $AP^{seg}_{50}$ | $AP^{seg}_{75}$ | $mAP^{box}$ | $AP^{box}_{50}$ | $AP^{box}_{75}$ |
|---|---|---|---|---|---|---|
| Baseline [22] | 23.7 | 41.5 | 23.3 | 29.2 | 52.9 | 29.3 |
| MBLLEN [41] | 22.5 | 40.7 | 22.3 | 28.5 | 52.0 | 28.4 |
| Zero-DCE [20] | 25.1 | 44.5 | 24.6 | 30.3 | 55.3 | 29.4 |
| DENet [46] | 16.1 | 31.0 | 15.4 | 19.5 | 40.0 | 15.7 |
| DarkIR [15] | 27.4 | 46.3 | 27.5 | 32.7 | 56.7 | 34.4 |
| YOLA [5] | 24.9 | 44.8 | 24.2 | 30.7 | 56.4 | 29.3 |
| FeatEnHancer [21] | _29.1_ | _48.7_ | _29.7_ | _34.0_ | _57.6_ | _35.3_ |
| **FRBNet(Ours)** | **30.2** | **50.5** | **30.4** | **36.9** | **61.2** | **38.4** |

Beyond the segmentation results already analyzed in the main text, this dataset also contains bounding box annotations for detection. Therefore, we conducted additional experiments and found: FRBNet shows even larger gains, achieving 36.9% $AP^{box}$ compared to 34.0% for FeatEnHancer and 32.7% for DarkIR. This indicates a 2.9-point improvement over the previous state-of-the-art. Notably, our approach demonstrates the most substantial improvement at $AP^{box}_{75}$ (38.4% vs. 35.3%), which requires more precise localization, highlighting the effectiveness of our frequency-domain features for accurate object boundary delineation.

## A.4 Extended Experiments

To further demonstrate the flexibility and task-level generalization of our frequency-domain feature enhancer, we additionally conducted experiments on two more tasks:

**FRBNet on Low-light Image Classification.** We adopt the official implementation of the low-light image classifier, using ResNet-101 as the backbone and following the standardized training protocols prescribed for the CODaN[30] dataset. For comparison, we include the Baseline model, FeatEnhancer [21], and YOLA [5] as representative prior approaches. The reported accuracies in Table10 demonstrate that our method consistently outperforms all competitors, achieving the highest classification accuracy among the evaluated methods. This performance gain can be attributed to our frequency-domain design, which effectively preserves discriminative cues under challenging low-light conditions.

**FRBNet on Low-light Video Action Recognition.** Furthermore, we conduct experiments on low-light video action recognition using the ARID dataset [67], implemented within the `MMAction2` framework and employing the TSN [61] architecture with a ResNet-50 backbone. We evaluate model performance using both Top-1 and Top-5 accuracy metrics, as reported in Table11, our method achieves substantial improvements over both the Baseline and YOLA [5], registering the highest scores across all metrics.

Table 10: Quantitative comparisons of the Low-light Image Classification.

| Method | Acc |
|---|---|
| Baseline [55] | 86.8 |
| FeatEnHancer [21] | 82.0 |
| YOLA [5] | 86.4 |
| **FRBNet(Ours)** | **88.2** |

Table 11: Quantitative comparisons of the Low-light Video Action Recognition.

| Method | Top-1 | Top-5 |
|---|---|---|
| Baseline | 42.65 | 96.27 |
| YOLA [5] | 41.09 | 93.68 |
| **FRBNet(Ours)** | **44.84** | **96.53** |

To further examine the effectiveness of our proposed FRBNet, we conducted additional ablation experiments by replacing the learnable frequency-domain filter with standard convolutional layers. Specifically, we implemented 3×3, 5×5, and 7×7 convolution kernels and evaluated these variants on the ExDark and DarkFace datasets using YOLOv3. As reported in the Table 12, our method consistently and significantly outperforms all convolution-based counterparts across different kernel sizes. This observation aligns with our original motivation: while standard convolutions exhibit spatial shift-invariance, they are inherently limited in capturing structured dependencies within the frequency domain. In contrast, our approach employs a radial basis function (RBF) network to construct a spectrally selective and directionally modulated filter, enabling the preservation and enhancement of localized frequency cues under real-world low-light conditions. Such spectral-directional adaptability is particularly crucial for effectively handling the non-uniform illumination of low-light imagery.

Table 12: Additional ablation experiments on FCR with convolution layers of varying kernel sizes.

| Method | ExDark | | DarkFace | |
|---|---|---|---|---|
| | Recall | mAP | Recall | mAP |
| Our | **90.6** | **74.9** | **75.7** | **57.7** |
| Conv 3×3 | 84.6 | 72.7 | 72.4 | 55.3 |
| Conv 5×5 | 85.4 | 73.0 | 73.7 | 54.8 |
| Conv 7×7 | 85.3 | 72.3 | 72.6 | 54.6 |

## A.5 Discussion

**Limitations.** Although FRBNet has demonstrated impressive performance in various low-light visual tasks, there still exist limitations. Due to the current design of FRBNet mainly addresses image degradation issues related to illumination, FRBNet may show less effective performance in low-light scenes with more complex degradation conditions, such as motion blur. This problem may be solved

by introducing all-in-one modules or forming models that consider more potential types of image degradation, which is a direction of our future work.

**Broader Impacts.** Our work contributes to enhancing visual perception systems in low-light environments, with potential applications in safety-critical domains like autonomous driving, surveillance, and emergency response. However, improved low-light vision perception also raises privacy concerns, as it might enable surveillance in previously invisible lighting conditions. We encourage responsible deployment of these technologies with appropriate privacy safeguards and regulatory compliance.

