# OpenReview forum: "FRBNet: Revisiting Low-Light Vision through Frequency-Domain Radial Basis Network"
_NeurIPS.cc/2025/Conference — NeurIPS 2025 poster_

### Official Review · Reviewer_tvFq · 2025-06-02

**Clarity:** 3
**Significance:** 3
**Originality:** 3
**Rating:** 4
**Confidence:** 4

**Summary:**

This paper revisits low-light image formation and extends the classical Lambertian model to better characterize low-light conditions. By shifting our analysis to the frequency domain, the authors theoretically prove that the frequency-domain channel ratio can be leveraged to extract illumination-invariant features via a structured filtering process. A plug-and-play module named FRBNet is proposed to be integrated into existing networks for low-light downstream tasks without modifying loss functions.

**Questions:**

See "Weakness".

**Ethical Concerns:**

["NO or VERY MINOR ethics concerns only"]

**Final Justification:**

Thanks for the rebuttal. I will keep my positive score.

**Limitations:**

See "Weakness".

**Paper Formatting Concerns:**

No.

**Quality:**

3

**Strengths And Weaknesses:**

* Strengths
  * The proposed method is the first work that operates the channel ratio for illumination-invariant features in the frequency domain.
  * The figures are very cool, and the performance is good.
* Weaknesses
  * The experimental results provide only the downstream tasks. Since this paper is not specific for "Object Detection and Beyond Under Low-Light Vision" as FeatEnHancer, the authors should also show the comparisons of the enhanced images with other methods (both numerical results and visual results, otherwise this paper is not comprehensive enough). Besides, the ablation study also provides only the downstream tasks.
    * Please provide the comparisons like DarkIR to show the enhanced images.
    * Please also evaluate the performance of low-light enhancement on  LOLBlur dataset [a],  Real-LOLBlur dataset [b], and  LOLv2-Real dataset [c] instead of only evaluate on the downstream applications.
      * [a] Lednet: Joint low-light enhancement and deblurring in the dark
      * [b] Real-world blur dataset for learning and benchmarking deblurring algorithms.
      * [c] Sparse gradient regularized deep retinex network for robust low-light image enhancement
  * The related work section should be put right after the introduction. Besides, please also include the discussions about the low-light enhancement methods such as [a,b,c].
  * It seems that the proposed method does not take the camera response function into consideration. The model requires that the camera have a linear response. I'm not sure whether the proposed method has a good generalization ability or not.
  * Does the proposed method take the dynamic range and pixel saturation into consideration?

---

> ### Author Rebuttal · Authors · 2025-07-30
>
> **Thank you very much for your positive feedback on our work!**  We have tried to address all the concerns in the following:
>
> ---
>
> > ***W1: The experimental results provide only the downstream tasks…***
>
> We appreciate your thoughtful comments and would like to clarify a potential misunderstanding. Our method is **not aimed at low-level image enhancement** in the low-light sense. Instead, following the motivation of FeatEnHancer and YOLA (which are both also our main comparison methods under PnP), our focus is on improving performance in low-light vision downstream tasks. Therefore, both our main comparisons and ablation studies are centered around detection and segmentation benchmarks.
>
> Our method is designed as a lightweight plug-and-play module for enhancing illumination-invariant features, and is not intended for perceptual restoration. As such, it is not directly suitable for evaluation on restoration benchmarks like LOL[1] or LOLv2[2], which focus on human-perceived image quality (pixel-level image quality evaluation indicators: PSNR, SSIM, etc.). We include visualizations in Figure 3(b). The outputs reflect machine-oriented representations before the backbone, not visually enhanced images(like DarkIR). Our model is trained end-to-end with downstream task losses, and its objective is to boost the quality of low-light visual understanding, rather than the appearance of the image itself. We thank you again for the constructive suggestions and encouragement.
>
> ---
>
> > ***W2: The related work section should be put right after the introduction. Besides, please also include the discussions about the low-light enhancement methods such as [a,b,c].***
>
> A2: Thank you for your concerns. We acknowledge that placing the related work section immediately after the introduction is a common and reasonable choice. However, in our case, this is a stylistic decision. We intended for readers to first follow the theoretical motivation and method design after the introduction. This structure is increasingly adopted in recent works, especially for theory-driven papers, and we believe it supports a more coherent flow for readers familiar with the domain. That said, **we fully respect your suggestion and will revise the Section order** accordingly in the final version to align with standard conventions and enhance clarity for a broader audience.
>
> Thank you for pointing out these related low-light enhancement works [a, b, c]. We acknowledge that our current paper may underrepresent this, largely due to page constraints and the different focus of our work. Our method is not designed to enhance image appearance, but instead aims to provide a PnP, task-driven solution for low-light vision downstream tasks. As for the suggested traditional enhancement methods and dataset: Retinex-based approaches [c], deblurring datasets like [b], and joint deblurring-enhancement frameworks such as LEDNet [a] are primarily designed to improve image visibility through pixel-level restoration or reconstruction. In contrast, our method doesn’t perform image enhancement, but focuses on learning illumination-invariant features in the frequency domain to support downstream tasks.
> > [a] Lednet: Joint low-light enhancement and deblurring in the dark.
> > [b] Real-world blur dataset for learning and benchmarking deblurring algorithms.
> > [c] Sparse gradient regularized deep retinex network for robust low-light image enhancement.
>
> Nevertheless, we agree that a broader context is valuable, and we will include discussion of these works in the *Related Work* Section.
>
> ---
>
> > ***W3: It seems that the proposed method does not take the camera response function into consideration. The model requires that the camera have a linear response. I'm not sure whether the proposed method has a good generalization ability or not.***
>
> A3: Thank you for pointing this out. While our method is inspired by the physical imaging model, we would like to clarify that our method doesn't strictly require a linear CRF. Instead of explicitly modeling the CRF, our method focuses on extracting discriminative, illumination-invariant features via end-to-end learning with downstream task-specific supervision. The network is trained directly on downstream objectives, which allows it to implicitly adapt to the statistical variations introduced by different CRFs in real-world data. Our experiments span multiple datasets(ExDark, DarkFace, ACDC, LIS) captured by diverse cameras, and our method demonstrates strong generalization performance across all of them. We agree that this is a meaningful aspect. We plan to explore integration with CRF techniques to improve performance in future work.
>
> ---
>
> >***W4: Does the proposed method take the dynamic range and pixel saturation into consideration?***
>
> A4: Thank you for this important concern. Similar to the concern of the Reviewer s1m7, dynamic range limitations and pixel saturation are indeed important challenges in low-light scenes.
> While our method does not explicitly model saturation effects, it is designed to improve structural robustness through a learnable frequency-domain filter, which helps preserve consistent patterns and suppress the influence of locally overexposed regions.
> To further evaluate the robustness of our method under pixel saturation, we conducted an additional experiment using the ExDark dataset. We applied a nonlinear gamma transformation to simulate partial overexposure under the Single, Weak, and Screen lighting conditions in ExDark. We then test our method, the baseline, and YOLA(*NeurIPS-2024*) on these transformed images. As shown in Table 1, our method maintains superior performance, confirming that it is more resilient to pixel saturation. We agree that incorporating explicit modeling of dynamic range and saturation effects is a valuable direction, and we consider this a promising avenue for future work.
>
> | Dataset | Method | Recall | mAP |
> | - | --- | --- | --- |
> | |Baseline | 88.7 | 78.7|
> | Exdark_var.   | YOLA (*NeurIPS2024*)| 89.3 | 79.9 |
> | | Our Method| **91.2** | **80.8** |
>
> **Table 1:** Quantitative comparisons based on YOLOv3 detector using ExDark_var.(Overexposure)
>
> ---
> ***Reference***
> > [1] Chen Wei et al. Deep Retinex Decomposition for Low-Light Enhancement. BMVC 2018.
>
> > [2] Wenhan Yang et al. Sparse Gradient Regularized Deep Retinex Network for Robust Low-Light Image Enhancement. TIP 2021.

---

### Official Review · Reviewer_cndh · 2025-07-01

**Clarity:** 3
**Significance:** 2
**Originality:** 3
**Rating:** 4
**Confidence:** 4

**Summary:**

This work proposes a frequency-domain radial basis network for object detection and semantic segmentation under low-light conditions. To address the incomplete modeling of low-light environments in existing methods, the authors extend the Lambertian model with the Phong illumination model to account for non-uniform lighting. They further introduce frequency-domain channel ratios and learnable frequency-domain filters to enhance illumination-invariant features. The proposed method is plug-and-play, and its effectiveness is supported by performance evaluations and algorithmic analysis.

**Questions:**

1. The authors introduce their plug-and-play solution by reviewing four mainstream strategies for low-light visual tasks, highlighting their own innovation. However, plug-and-play approaches also share a notable limitation: they often lack sufficient interaction with the backbone of downstream perception networks. In essence, these methods still resemble preprocessing approaches, albeit operating in the feature domain. I encourage the authors to explicitly discuss this issue.

2. Given the newly proposed modeling, it remains questionable whether illumination, reflection, and specular components can be effectively separated in the frequency domain. The authors are encouraged to provide further feature-level analysis to strengthen the theoretical justification.

3. Since the proposed modeling is intended to address the incomplete treatment of low-light environments in prior work, it would be beneficial to provide more comprehensive feature-level analysis under highly non-uniform lighting conditions—e.g., nighttime scenes with localized lighting such as street lamps.

4. On the downstream task side, the work only considers detection and segmentation. It would be valuable to extend the range of downstream tasks to better validate the flexibility and generalizability of the proposed method. Additionally, as the method essentially functions as a feature enhancer trained solely under task-specific losses, further analysis is needed to assess its interpretability and whether the claimed benefits are indeed justified.

5. The performance evaluation lacks comparison with representative joint learning methods, such as MAET, IAT, DANNet, and GLASS-GPS. Including such comparisons would provide a more complete assessment of the method’s effectiveness.

**Ethical Concerns:**

["NO or VERY MINOR ethics concerns only"]

**Final Justification:**

After carefully reviewing the authors’ rebuttal, I believe they have partially addressed my earlier concerns. Considering this, along with the perspectives of the other reviewers, I have decided to raise my rating to borderline accept.

**Limitations:**

Yes

**Quality:**

3

**Strengths And Weaknesses:**

The motivation of the proposed method is well-articulated, with a clear presentation of the overall approach and theoretical derivation. The experiments are relatively comprehensive. However, the analysis of existing plug-and-play methods could be further strengthened. One key limitation is that most such methods still focus on feature-level preprocessing, with limited interaction with the backbone of downstream tasks. Additional evidence is needed to support whether components such as illumination, reflection, and highlight residuals can be effectively separated in the frequency domain. The numerical stability of the input preprocessing also requires clarification. Moreover, the experimental section appears to lack comparisons with joint learning approaches.

---

> ### Author Rebuttal · Authors · 2025-07-30
>
> **We sincerely appreciate the reviewer’s constructive feedback.** Below are detailed responses to your questions:
> > ***Q1: The authors introduce their plug-and-play solution by reviewing four mainstream strategies for low-light visual tasks, highlighting their own innovation. However, plug-and-play approaches also share a notable limitation: they often lack sufficient interaction with the backbone of downstream perception networks. In essence, these methods still resemble preprocessing approaches, albeit operating in the feature domain. I encourage the authors to explicitly discuss this issue.***
>
> A1: Thank you for this thoughtful comment. We acknowledge that plug-and-play approaches often face the limitation of limited interaction with downstream backbones. However, as recognized by **Reviewer s1m7**, the PnP strategy provides adaptability, with both efficiency and performance. We also believe its training efficiency and modularity make it highly practical for real-world deployment.
>
> Importantly, our method differs from classical preprocessing approaches. As noted in *Line 129*, image enhancement methods typically optimize for human visual quality but may not guarantee machine perception performance. In contrast, our method is trained end-to-end with downstream task-specific supervision, aiming to extract robust illumination-invariant features that directly benefit downstream tasks, rather than merely improving the visual appearance of images. This strategy preserves the modular advantages and ensures tight alignment with downstream task goals.
>
> ---
> > ***Q2: Given the newly proposed modeling, it remains questionable whether illumination, reflection, and specular components can be effectively separated in the frequency domain. The authors are encouraged to provide further feature-level analysis to strengthen the theoretical justification.***
>
> A2: Thank you for raising this point.
> Based on the extended generalized low-light modeling and the theoretical analysis in Section 2, our newly proposed frequency-domain channel ratio (FCR) is composed of illumination, reflection, and high-lit residual (evolved from the specular).
> Due to being converted to the frequency domain, these components exhibit different characteristics due to their respective natures.
> - The illumination component usually exhibits smooth fluctuations within the low-frequency spectrum.
> - The reflection component is usually stable and represents the material and detailed information of the object, contributing to mid-to-high frequency responses.
> - The high-lit residual component, which is derived from the specular term with sharp local peaks and directionally biased energy, usually manifests as localized specific orientations in the frequency domain.
>
> Therefore, **our proposed learnable filter aims to enhance illumination invariant features for low-light recognition by suppressing fluctuating illumination interference and modulating the high-lit residual component, rather than explicitly separating these three components.** In Figure 3, we have already provided feature-level visualizations, including both feature maps and heatmaps, to demonstrate the effective enhancement of our method for illumination-invariant features. Following your suggestion, we will include additional visual and spectral analyses in the Appendix.
>
> ---
> > ***Q3: Since the proposed modeling is intended to address the incomplete treatment of low-light environments in prior work, it would be beneficial to provide more comprehensive feature-level analysis under highly non-uniform lighting conditions—e.g., nighttime scenes with localized lighting such as street lamps.***
>
> A3: Thank you for the helpful suggestion. We will add the feature-level analysis under highly non-uniform lighting conditions in Appendix A.3. Our method is motivated by the need to handle highly non-uniform lighting conditions. To this end, we propose the corresponding modeling and an effective method to enhance illumination invariant features for the non-uniform lighting conditions in low-light vision. To validate the effectiveness of the proposed method, we conduct experiments on challenging datasets, including the ACDC-night dataset.
> **The ACDC-night dataset contains diverse real-world scenarios with localized lighting sources like vehicle headlights and street lamps**.
>
> The visualized segmentation results of the ACDC-night dataset have been provided in Appendix A.3 (Fig.7). In nighttime scenes with localized illumination, our proposed method can extract more robust illumination invariant features without being affected by localized lighting interference for better category recognition and edge preservation thanks to the extended modeling and the effective learnable frequency-domain filter. Furthermore, we will include feature map comparisons under such localized lighting conditions to better illustrate how the model responds in these cases.
>
>
> ---
> > ***Q4: On the downstream task side, the work only considers detection and segmentation. It would be valuable to extend the range of downstream tasks to better validate the flexibility and generalizability of the proposed method. Additionally, as the method essentially functions as a feature enhancer trained solely under task-specific losses, further analysis is needed to assess its interpretability and whether the claimed benefits are indeed justified.***
>
> A4: Thank you for your concern. We would like to clarify that our method has already been evaluated across four downstream tasks using four distinct frameworks and four different datasets:
> &emsp; a. Object Detection (YOLOv3 on ExDark)
> &emsp; b. Face Detection (TOOD on DarkFace)
> &emsp; c. Semantic Segmentation (DeepLabV3+ on ACDC-Night)
> &emsp; d. Instance Segmentation (Mask R-CNN on LIS)
>
> To further demonstrate the flexibility and task-level generalization of our frequency-domain feature enhancer, we additionally conducted experiments on two more tasks:
> - **Low-light Image Classification** on the CODaN[1] dataset using ResNet-101 as backbone. We compare our method against baseline, FeatEnhancer, and YOLA. As shown in Table 1, our method achieves the best accuracy among all methods(Tab.1).
> - **Low-light Video Action Recognition** on the ARID[2] dataset using the TSN[3] framework (ResNet-50 backbone). Evaluated with Top-1 and Top-5 accuracy, our method again outperforms the baseline significantly(Tab.2).
>
> These results show that our method can be plugged into diverse vision pipelines.
>
> |Method| Acc |
> | - | - |
> | Baseline | 86.8 |
> | FeatEnhancer | 82.0 |
> | YOLA | 86.4 |
> | Our | **88.2** |
>
> **Table 1:**  Quantitative comparisons of low-light image classification.
>
> |Method|Top-1|Top-5|
> |-|-|-|
> |Baseline|42.65|96.27|
> |Our| **44.84**|**96.53**|
>
> **Table 2:**  Quantitative comparisons of low-light video action recognition.
>
> Regarding **interpretability**, while our method is trained with task-specific losses, it is guided by a physically motivated formulation (Sec.2) that separates illumination and reflectance components in the frequency domain. This formulation leads to two specific design choices: a zero-DC frequency window to suppress global illumination shifts, and a radial basis filter with directional modulation to preserve meaningful structural cues. We will include the above analysis and additional experimental results in the Appendix.
>
> ---
> > ***Q5: The performance evaluation lacks comparison with representative joint learning methods, such as MAET, IAT, DANNet, and GLASS-GPS. Including such comparisons would provide a more complete assessment of the method’s effectiveness.***
>
> A5: We appreciate your emphasis on a comprehensive assessment. As shown in Tab.1, the representative joint learning methods **including MAET and IAT**, are categorized under the "Multi-task" learning paradigm, and **their results are presented**.
>
> Regarding DANNet[4] and GLASS-GPS[5] (perhaps referring to "GPS-GLASS"?), they are both **unsupervised domain adaptation(UDA)** methods, which is different from our supervised learning research. More specifically, DANNet[4] leverages adversarial learning using paired day-night images and pseudo labels from daytime predictions, while GPS-GLASS[5] depends on GPS-aligned image pairs and optical flow from daytime videos to provide pseudo-supervision. Both frameworks are highly specialized for semantic segmentation and rely on specific data, such as scene alignment paired data and optical flow matching, so they are not available in our work.
>
> For the sake of representativeness and fairness in comparison, we provide the results of methods in four paradigms in low-light object detection in Tab.1. Furthermore, the results of semantic segmentation, face detection, and instance segmentation in low-light vision are presented in Tab.2, Fig.6, and Fig.8. In comparison with a wide range of SOTA methods, our method consistently achieves the best results.
>
> ---
> **Reference**
> > [1] Attila Lengyel et al. Zero-Shot Domain Adaptation with a Physics Prior. ICCV 2021.
>
> > [2] Yuecong Xu et al. ARID: A New Dataset for Recognizing Action in the Dark. DL-HAR 2021.
>
> > [3] Limin Wang et al. Temporal Segment Networks: Towards Good Practices for Deep Action Recognition. ECCV 2016.
>
> > [4] Xinyi Wu et al. DANNet: A One-Stage Domain Adaptation Network for Unsupervised Nighttime Semantic Segmentation. CVPR 2021.
>
> > [5] Hongjae Lee et al. GPS-GLASS: Learning Nighttime Semantic Segmentation Using Daytime Video and GPS data. ICCVW 2023.

---

> > ### Comment · Reviewer_cndh · 2025-08-05
> >
> > After carefully reviewing the authors’ rebuttal, I believe they have partially addressed my earlier concerns. Considering this, along with the perspectives of the other reviewers, I have decided to raise my rating to borderline accept.

---

> > > ### Author Response · Authors · 2025-08-06
> > >
> > > **We are deeply grateful to you for acknowledging our work after reviewing our rebuttal !**
> > > We formally commit to thoroughly revising the paper for the final version, incorporating all of your valuable feedback to enhance its quality.
> > >
> > > Thank you again for your time, effort, and insightful comments on our paper :)

---

### Official Review · Reviewer_2mko · 2025-07-01

**Clarity:** 2
**Significance:** 2
**Originality:** 3
**Rating:** 4
**Confidence:** 4

**Summary:**

This paper proposes an image processing module for computer vision tasks under low-light conditions. The module can be trained using only downstream task losses (plug-and-play).

The authors extend the traditional Lambertian model (which only considers diffusion components) by adding local light source terms from the Phong model. By converting color ratios to log scale and transforming them to frequency space, they demonstrate that highlight components can be linearly separated.

In the proposed method, they apply a learnable frequency domain filter (LFF) to the Fourier-transformed log-scaled color ratios. After applying LFF and performing inverse Fourier transformation, they concatenate the resulting log-scale color ratios and process them with convolution. Similarly, they apply convolution to the original image, concatenate both the processed color ratio and original image, and apply another convolution before feeding the result into CV tasks.
They show the method improve multiple computer vision tasks under low light conditions.

**Questions:**

1. Since $F_{inv}^{C_i C_j}$ corresponds to $log(I_{C_i}/I_{C_j})$, is there a specific reason for concatenating with the original image I rather than transforming back via exp? What is the reason for concatenating instead?

2. Is the number of channels in $F_{out}$ equal to 3? If so, is the result image displaying $F_{out}$? There doesn't seem to be any visible difference from the GT image.

3. How were the predefined $\mu_k$ values determined?

4. Would it be possible to conduct an ablation study using only FCR with Convolution layers?

**Ethical Concerns:**

["NO or VERY MINOR ethics concerns only"]

**Final Justification:**

The rebuttal explanations address my main concerns, and the authors have agreed to implement the requested revisions.
I have therefore updated my recommendation accordingly.

**Limitations:**

yes

**Paper Formatting Concerns:**

No major formatting issues detected.

**Quality:**

2

**Strengths And Weaknesses:**

Strengths:
Originality:
- Proposal of a novel approach:
  - Extension of the Lambertian model with the Phong model, implementing filtering in frequency space
  - Learning method using downstream task losses (plug and play)
  - Application of knowledge about separating local light sources to preprocessing for CV tasks

Weaknesses:
Quality:
- Theoretical foundation:
  - Unclear validity of the assumption that $H_C$ is very small compared to ambient light. It seems to have much higher irradiance than diffuse components.
  - The derivation process of equation (8) is not explicitly shown, and the rationale for assuming color channels have intensity only at specific frequencies is unclear

Clarity:
- Explanation of the method is unclear and difficult to understand:
  - Please add an explanation of the Radial basis function, particularly how the predefined $\mu_k$ values were determined
  - The number of channels in $F_{out}$ and the method of generating the final output (details of Convolution layer) are not clear
- Weak connection between theoretical background and proposed method:
  - Insufficient explanation of how filtering in frequency space solves the local light source problem
  - The relationship between equations (8)-(10) and LLF (spatial Gaussian, Gaussian weights for values, weights in frequency space) is not clear

Significance:
- This method requires to prepare low light data and labels for each task for training

Minor comment:
In Section 2, the statement "As shown in Eq. (5), the nonlinear residual from the highlight term disrupts the clean separation of illumination and reflectance, limiting the effectiveness of spatial-domain channel ratio methods" - shouldn't this be "separation of highlight and diffuse" rather than "separation of illumination and reflectance"?

---

> ### Author Rebuttal · Authors · 2025-07-30
>
> **Thank you for your constructive assessments of our work,** and we address the concerns point by point as follows:
>
> > ***W1: Unclear validity of the assumption that $H_C$ is very small compared to ambient light. It seems to have much higher irradiance than diffuse components.***
>
> A1: Thank you for this comment. We respectfully clarify that we **don’t assume** $H_C$ is very small compared to ambient light in terms of irradiance. $H_C$ is not absolute irradiance, but a coefficient. As stated in Section 2.1, we define $H_C$ as the relative intensity between $D_C$ (the spatially irregular highlight componet) and $S_C$ (the standard diffuse reflection componet). The relationship of them is  $ S_C (x,y)=H_C (x,y) \cdot D_C (x,y) $ and $H_C (x,y)∈[0,1)$. We will further clarify this to avoid potential misinterpretation in the revised version.
>
> ---
> > ***W2: The derivation process of equation (8) is not explicitly shown, and the rationale for assuming color channels have intensity only at specific frequencies is unclear.***
>
> A2: Thank you for your insights into the theoretical foundation. We would like to clarify that equation (8) is a standard polar-form representation in complex space, where each frequency component can be expressed as an amplitude-phase pair:
> $\mathcal{H}=a \cdot e^{i \theta}$. This is commonly used in frequency-domain analysis, and we have provided a more detailed explanation in Appendix A.2 (Eqs. 22-24).
>
> We also would like to clarify that we **don’t assume** color channels have intensity only at specific frequencies. Our analysis is conducted across the entire frequency spectrum, not limited to specific bands. We will consider your concerns and make modifications to the manuscript to make it clearer.
>
> ---
> > ***W3&Q3: Please add an explanation of the Radial basis function, particularly how the predefined $\mu_{k}$ values were determined. & How were the predefined $\mu_{k}$ values determined?***
>
> A3: Thank you for your suggestion. We will include a more detailed explanation of the radial basis function in the Tech. Appendix. We employ nonlinear Gaussian radial basis functions to construct a learnable filter that captures responses at different frequencies.
>
> The predefined $\mu_{k}$∈[0,1] values are determined according to the well-established convention in Gaussian kernel radial basis function networks to represent normalized radial frequency radii which are selected to be evenly spaced across this range.
>
> ---
> > ***W4&Q2: The number of channels in $F_{out}$ and the method of generating the final output (details of Convolution layer) are not clear. & Is the number of channels in $F_{out}$ equal to 3?***
>
> A4: Thank you for your question. As illustrated in Figure 2 (bottom-left corner), the number of channels in $F_{out}$ is 3. The details of the convolution layer (structure, channel dimensions, etc.) are also provided there. We apologize for the inconvenience caused by the small font in Figure 2 and will optimize the display of this part in the revised version. In addition, we will explicitly describle the convolutional structure to avoid ambiguity. More comprehensive implementation details are also available in our released code.
>
> ---
> > ***Q2: If so, is the result image displaying $F_{out}$? There doesn't seem to be any visible difference from the GT image.***
>
> A4(2): We would like to clarify the distinction between various result images. Figure 3(a) presents **the final results of downstream tasks**(e.g. bbox and mask) after processing by the entire pipeline. Compared to other methods, our results exhibit clearer boundaries and more accurate bounding boxes, which are closer to the ground truth. Figure 3(b) presents the visualization of extracted invariant features $F_{out}$ from different plug-and-play modules including FeatEnHancer, YOLA, and the proposed method. **The similarity between our result images and the GT images confirms the effectiveness of the proposed method**. We will further clarify the results shown in Figure 3 to avoid misinterpretation.
>
> ---
> > ***W5: Insufficient explanation of how filtering in frequency space solves the local light source problem.***
>
> A5: Thank you for highlighting this point. We will expand the explanation in the revised version to clarify how our method addresses the influence of local light sources. Local light source appears as sharp, localized brightness variations in the spatial domain, which correspond to structured high-frequency components in the frequency domain. As established in [1,2], the low-frequency components are often associated with smooth illumination variations and high-frequency components with reflectance edges and details. Our method leverages this property by applying learnable frequency-domain filters to select inconsistent frequency signals. As discussed at the end of Section 2.2:
>
> `we design specialized filtering strategies that aim to robustly extract invariant illumination features thus enhancing the reliability and effectiveness of feature extraction under varying lighting conditions.`
>
> We will make this more explicit in the revised version.
>
> ---
> > ***W6: The relationship between equations (8)-(10) and LLF (spatial Gaussian, Gaussian weights for values, weights in frequency space) is not clear.***
>
> A6: We respectfully clarify that terms such as “LLF”, “spatial Gaussian”, “Gaussian weights for values”, and “weights in frequency space” **do not appear** in our paper, which leads to some confusion on our side. We infer the reviewer may seek a connection between the theoretical equations and our method. This connection is described at the beginning of Section 3, where we state that our method is inspired by extended image formation principles. Specifically, Eqs. (8)–(10) present the derivation of our frequency-domain channel ratio, which forms the theoretical foundation for our frequency-domain radial basis network in Section 3.
> We will revise the paper to avoid misinterpretation.
>
> > ***W7: This method requires to prepare low light data and labels for each task for training***
>
> A7: Thank you for this significant comment. Our method is designed to improve task performance under realistic illumination constraints in low-light vision by **supervised learning**, where training with labeled low-light data is common and necessary. Our method is implemented as a plug-and-play feature extraction module, which can be integrated into existing networks without changes to the backbone. We appreciate your concern and plan to explore unsupervised extensions in future works.
>
> ---
> >***Q1:Since $F_{inv}^{C_i C_j}$ corresponds to $log (I_{c_i} / I_{c_j})$, is there a specific reason for concatenating with the original image I rather than transforming back via exp? What is the reason for concatenating instead?***
>
> A8: We would like to clarify that $F_{inv}^{C_i C_j}$ is **not simply corresponding to** $log (I_{c_i} / I_{c_j})$, but rather the result of nonlinear learnable frequency-domain filtering (LFF) applied to log-ratio inputs. Our goal is not to reconstruct the image, but to extract illumination-invariant representations. The log-ratio is a well-established technique for enhancing structural contrast. Applying exp would simply reverse the transformation, potentially losing the beneficial properties of the log space. We concatenate the illumination-invariant feature with the original image feature to combine complementary information. This fusion enables the backbone to exploit both robustness and expressiveness, and is also supported by findings in related works[3].
>
> ---
> > ***Q4: Would it be possible to conduct an ablation study using only FCR with Convolution layers?***
>
> A9: Thank you for the suggestion. We conduct an additional experiment by replacing our learnable frequency-domain filter with standard convolution layers of kernel sizes 3×3, 5×5, and 7×7, forming a baseline of “FCR with Convolution Layers”. The results are reported in Table 1, where our method significantly outperforms all convolution-based variants on both ExDark and DarkFace datasets, particularly in Recall and mAP.
>
> | |ExDark| |DarkFace| |
> |-|-|-|-|-|
> | |Recall|mAP|Recall|mAP
> |Our|**90.6**|**74.9**|**75.7**|**57.7**
> |Conv 3*3|84.6|72.7|72.4|55.3
> |Conv 5*5|85.4|73.0|73.7|54.8
> |Conv 7*7|85.3|72.3|72.6|54.6
>
> **Table 1:**  Ablation study on FCR with convolution layers of varying kernel sizes.
>
> This result reflects our original design motivation: **standard convolutions are spatially shift-invariant, but they are not well suited for modeling structured dependencies in the frequency domain**. In contrast, our approach leverages a radial basis function network to construct a spectrally selective and directionally modulated filter, which is critical for capturing localized frequency cues under real low-light conditions. We will include this ablation and a more detailed analysis in the revise version.
>
> ---
> > ***Response to Minor Comment***
>
> A10: Thank you for the suggestion. In Section 2.1, we have defined the illumination and reflectance components based on our extended image formation model. Since the “diffuse” term is traditionally embedded within the Lambertian formulation (which includes both illumination and reflectance), we chose to describe our objective in terms of isolating illumination-invariant features by separating the illumination and reflectance. Therefore, we believe keeping the current terminology helps maintain consistency and clarity. We appreciate your careful reading and the helpful minor comment :)
>
> ---
> **Reference**
> > [1] Haohan Wang et al. High-frequency component helps explain the generalization of convolutional neural networks. CVPR 2020.
>
> > [2] Yang Wang et al. Decoupling-and-aggregating for image exposure correction. CVPR 2023.
>
> > [3] Mingbo Hong et al. You Only Look Around: Learning Illumination-Invariant Feature for Low-light Object Detection. NeurIPS 2024.

---

> > ### Comment · Reviewer_2mko · 2025-08-04
> >
> > Thank you for the detailed rebuttal. The additional clarifications and the new ablation results address my main concerns. I will raise my recommendation to a Borderline Accept on the condition that:
> >
> > - Supplementary explanations from the rebuttal are integrated into the paper.
> >
> > - The new ablation study is added to the main manuscript or appendix.

---

> > > ### Author Response · Authors · 2025-08-04
> > >
> > > Thank you very much for your positive feedback! We sincerely appreciate your valuable suggestions. **We commit to revising the paper strictly in accordance with these suggestions:** we will integrate supplementary explanations into the paper to ensure clarity and completeness, and add the new ablation study to the appendix as specified.
> > >
> > > We will make every effort to improve the paper and ensure these revisions are thoroughly implemented. Thank you again for your guidance and support :)

---

### Official Review · Reviewer_s1m7 · 2025-07-04

**Clarity:** 3
**Significance:** 3
**Originality:** 3
**Rating:** 4
**Confidence:** 3

**Summary:**

The authors present a novel frequency-domain framework for extracting illumination invariant features in dark images by learning radial basis filters. This plug-and-play module can be seamlessly integrated into existing low-light vision networks and achieves performance improvements.

**Questions:**

The same as in the weakness points.

**Ethical Concerns:**

["NO or VERY MINOR ethics concerns only"]

**Limitations:**

yes

**Paper Formatting Concerns:**

looks ok.

**Quality:**

3

**Strengths And Weaknesses:**

Strengths
The approach is not a totally data-driven technique and has a physical ground instead.
The PnP strategy provides adaptability.
The efficiency and performance are good.

Weaknesses
1. The overexposure and underexposure happen frequently in nighttime videos and violate the physical model. will this hamper the performance? why?
2. The relation to existing frequency-domain related work?
3. The segmentation performance on some classes is not that advantageous (some columns in Tab. 2), why?
4. I expect some failure examples and analysis.
5. The visualized features in Fig. 3(c) do not show advantages, those in the backbone look similar (signs might be opposite), and the heatmaps are not better (higher attention on the head is reasonable, but on corners make no sense).
6. The title is too big and I suggest focusing on the proposed technique.

---

> ### Author Rebuttal · Authors · 2025-07-30
>
> **Thank you for your constructive feedback and for recognizing the strengths of our work.** Below are detailed responses to your concerns:
> > ***W1: The overexposure and underexposure happen frequently in nighttime videos and violate the physical model. will this hamper the performance? why?***
>
> A1: Thank you for raising this important concern. Overexposure (pixel saturation leading to loss of detail) and underexposure are common challenges in low-light scenes. However, our approach does not aim to reconstruct raw pixels. We focus on extracting robust illumination-invariant features to support downstream tasks under real-world low-light conditions. To better model challenging lighting, we extend the classical Lambertian model by introducing a highlight term. Our method is designed to suppress interference and indirectly reduce the impact of overexposed regions on feature representation from the frequency domain. Importantly, our model is not intended as a precise law for all conditions, but rather as a generalized approximation that guides network design. Compared to purely data-driven methods, this physical grounding improves interpretability and robustness.
>
> To further validate this, we simulate overexposure using ExDark[1] data under “Single”, “Weak”, and “Screen” lighting conditions. We apply a gamma-based nonlinear transformation to create partially overexposed images. Then we evaluate our method, baseline models, and YOLA(NeurIPS24)[2] on these transformed images. This suggests that overexposure doesn’t significantly hamper the performance of our method.
>
> | Dataset | Method | Recall | mAP |
> | - | --- | --- | --- |
> | |Baseline | 88.7 | 78.7|
> | Exdark_var. | YOLA (*NeurIPS2024*)| 89.3 | 79.9 |
> | | Our Method| **91.2** | **80.8** |
>
> **Table 1:** Quantitative comparisons based on YOLOv3 detector using ExDark_var.(Overexposure)
>
> ---
> > ***W2: The relation to existing frequency-domain related work?***
>
> A2: We appreciate the opportunity to clarify this point. As discussed in Section 5.2, most existing frequency-domain methods primarily focus on image enhancement via spectral decomposition, typically modifying low-frequency illumination components while preserving high-frequency details[3,4,5]. In contrast, as **Reviewer tvFq** pointed out, our work is the first to adopt the channel ratio for illumination-invariant features in the frequency domain. This is a shift from pixel-level enhancement to feature-level learning. Furthermore, our angular-modulated radial basis filtering uniquely adapts to phase-structured interference, which is unexplored in prior works. We will revise the *Related Work* section to better analyze the relation to existing frequency-domain related work.
>
> ---
> > ***W3: The segmentation performance on some classes is not that advantageous (some columns in Tab. 2), why?***
>
> A3: Thank you for the detailed observation. While current results reveal category-specific challenges, the optimization objective of our method focuses on illumination invariant filtering to facilitate low-light downstream tasks universally. In semantic segmentation tasks, class imbalance and intra-class variance will lead to a decrease in segmentation accuracy for individual categories. For example, *terrain* and *pole* account for only 0.81% and 1.01% of total labels. Some classes (such as *terrain vs. vegetation*) exhibit highly similar textures, color cues, and blurred structural boundaries, especially under limited illumination. These factors undermine the discriminative features required for accurate segmentation, making such distinctions inherently difficult for all methods. Crucially, our method still delivers significant overall gains for low-light semantic segmentation (in Tab.2), addressing the core goal of low-light vision tasks. Furthermore, our method’s plug-and-play characteristic enables seamless integration with category-sensitive mechanisms (e.g., re-weighting losses), which will be a scalable and sustainable research direction in future work.
>
> ---
> > ***W4: I expect some failure examples and analysis.***
>
> A4: We agree that including failure cases and their detailed analysis would provide a more complete understanding of our work. Due to page limitations, we discussed the limitations of our method in Appendix A.4. As an example, in the last image of Fig. 7 (bottom row), our method shows imprecise segmentation on the lower right corner (confusing the fine contours of the tree markings with vegetation). This is likely due to motion blur in the input image caused by vehicle movement under low light, which suppresses high-frequency structural cues. While our method improves overall performance, such blur-related degradation remains a challenge.
> We will incorporate concrete failure examples and a more thorough analysis in the Appendix of the revised version.
>
> ---
> > ***W5:The visualized features in Fig. 3(c) do not show advantages, those in the backbone look similar (signs might be opposite), and the heatmaps are not better (higher attention on the head is reasonable, but on corners make no sense).***
>
> A5: This is an impactful review for improving our paper!  We will revise the paper to provide more details in Fig.3.
> The visualized feature map of “Baseline_backbone” is from the first layer of backbone with the input of the raw image, while the visualized feature map of “FRBNet_backbone” is from the same layer of the same backbone with the input processed by our method.
> Their overall structure may look similar because they are from the same backbone with the same input image. Our method enhances object details and reveals richer gradients, while the Baseline feature appears flatter and lacks spatial focus.
>
> The heatmaps are from the first layer’s output features of the neck part. Apart from corners, our method yields sharper responses around meaningful regions, such as the contours of the bicycle and the human head. More importantly, our method achieves the accurate detection of this image, while the baseline does not. The enhanced attention on the key semantic region supports the effectiveness of our method in improving low-light downstream tasks' performance in detection and segmentation.
> We will replace the visualization with a clearer version and include more examples in the Tech. Appendix.
>
> ---
> > ***W6:The title is too big and I suggest focusing on the proposed technique***
>
> A6: Thank you for the helpful suggestion. We will revise the title to: “FCRFilter: Learning Provable Illumination Invariance for Low-light Vision via Frequency-domain Channel Ratio Filtering”. Correspondingly, we will also revise the related terminology and abbreviations throughout the paper to maintain consistency.
>
> ***
> **Reference**
> > [1] Yuen Peng Loh et al. Getting to know low-light images with the Exclusively Dark dataset. CVIU 2019.
>
> > [2] Mingbo Hong et al. You Only Look Around: Learning Illumination-Invariant Feature for Low-light Object Detection. NeurIPS 2024.
>
> > [3] Wenbin Zou et al. Joint wavelet sub-bands guided network for single image super-resolution. IEEE Transactions on Multimedia 2022.
>
> > [4] Haohan Wang et al. High-frequency component helps explain the generalization of convolutional neural networks. CVPR 2020.
>
> > [5]  Chenxi Wang et al. Fourllie: Boosting low-light image enhancement by fourier frequency information. MM 2023.

---

### Note · Authors · 2025-08-14

We sincerely thank all *Four Reviewers* for their time, effort, and constructive feedback, as well as their positive ratings for our paper. We truly appreciate their insightful suggestions and comments, which are invaluable improving the quality of our paper.

We are encouraged by the positive feedback from the reviewers: **Reviewer s1m7** valued that our method is not purely data-driven but grounded in physical principles, incorporates a versatile PnP strategy, and demonstrates both strong efficiency and high performance. **Reviewer 2mko** recognized the originality of our work, highlighting the proposal of a novel approach. **Reviewer cndh** appreciated that the motivation of our method is well-articulated, with a clear and rigorous presentation of the overall approach and theoretical derivation. **Reviewer tvFq** noted that our method is the first to operate the channel ratio for illumination-invariant features in the frequency domain, commended the visual quality of our figures, and acknowledged the strong performance achieved.

During the rebuttal phase, we carefully addressed each reviewer’s comments point by point and resolved most of their concerns. In particular, we conducted additional experiments to evaluate our method under overexposure conditions, explore more diverse low-light downstream tasks, and provide further ablation studies. We also clarified technical details of the derivation and addressed misunderstandings.

For the final version, we are committed to incorporating all constructive suggestions, refining the experimental content, improving the related work section, and updating the references accordingly. Once again, we sincerely thank all reviewers for their valuable time and insightful feedback, which have been instrumental in strengthening our paper!

---

### Decision · Program_Chairs · 2025-09-17

**Decision:**

Accept (poster)

**Comment:**

This paper received four borderline, but positive leaning reviews, with all four reviewers converging to borderline accept recommendations (4 BA).

There was general appreciation for the novelty, motivation and physical grounding of the solution.

While there were some concerns raised in the original reviews about the quality of the results and the overall presentation, they were addressed to a large extent during the author-reviewer discussion phase. As a result, an accept decision was reached.

The authors are encouraged to account for the reviews while preparing the final camera-ready version.